# Identical twins carry a persistent epigenetic signature of early genome programming

Jenny van Dongen [1,2,3✉], Scott D. Gordon [4], Allan F. McRae [5], Veronika V. Odintsova [1,2,3], Hamdi Mbarek [1,2,3], Charles E. Breeze [6], Karen Sugden[7], Sara Lundgren[8], Juan E. Castillo-Fernandez [9], Eilis Hannon [10], Terrie E. Moffitt[7,11], Fiona A. Hagenbeek [1,3], Catharina E. M. van Beijsterveldt [1,3], Jouke Jan Hottenga [1,3], Pei-Chien Tsai[9], BIOS Consortium*, Genetics of DNA Methylation Consortium*, Josine L. Min [12,13], Gibran Hemani [12,13], Erik A. Ehli [14], Franziska Paul [15], Claudio D. Stern [16], Bastiaan T. Heijmans [17], P. Eline Slagboom [17], Lucia Daxinger [18], Silvère M. van der Maarel [18], Eco J. C. de Geus [1,3], Gonneke Willemsen[1,3], Grant W. Montgomery [5], Bruno Reversade [15,19,20], Miina Ollikainen [8], Jaakko Kaprio [8], Tim D. Spector [9], Jordana T. Bell [9], Jonathan Mill [10], Avshalom Caspi[7,11], Nicholas G. Martin [4] & Dorret I. Boomsma [1,2,3]

Monozygotic (MZ) twins and higher-order multiples arise when a zygote splits during pre-implantation stages of development. The mechanisms underpinning this event have remained a mystery. Because MZ twinning rarely runs in families, the leading hypothesis is that it occurs at random. Here, we show that MZ twinning is strongly associated with a stable DNA methylation signature in adult somatic tissues. This signature spans regions near telomeres and centromeres, Polycomb-repressed regions and heterochromatin, genes involved in cell-adhesion, WNT signaling, cell fate, and putative human metastable epialleles. Our study also demonstrates a never-anticipated corollary: because identical twins keep a lifelong molecular signature, we can retrospectively diagnose if a person was conceived as monozygotic twin.

[1] Department of Biological Psychology, Vrije Universiteit Amsterdam, Amsterdam, The Netherlands. [2] Amsterdam Reproduction and Development (AR&D) Research Institute, Amsterdam, The Netherlands. [3] Amsterdam Public Health Research Institute, Amsterdam, The Netherlands. [4] Queensland Institute of Medical Research Berghofer, Brisbane, QLD, Australia. [5] Institute for Molecular Bioscience, The University of Queensland, Brisbane, QLD, Australia. [6] Altius Institute for Biomedical Sciences, Seattle, WA, USA. [7] Department of Psychology and Neuroscience and Center for Genomic and Computational Biology, Duke University, Durham, NC, USA. [8] Institute for Molecular Medicine Finland FIMM, University of Helsinki, Helsinki, Finland. [9] Department of Twin Research and Genetic Epidemiology, Kings College London, London, UK. [10] University of Exeter Medical School, University of Exeter, Exeter, UK. [11] Institute of Psychiatry, Psychology & Neuroscience, King's College London, London, UK. [12] MRC Integrative Epidemiology Unit at the University of Bristol, Bristol, UK. [13] Population Health Science, Bristol Medical School, University of Bristol, Bristol, UK. [14] Avera Institute for Human Genetics, Sioux Falls, SD, USA. [15] Institute of Molecular and Cellular Biology, A*STAR, Singapore, Singapore. [16] Department of Cell and Developmental Biology, University College London, London, UK. [17] Molecular Epidemiology, Department of Biomedical Data Sciences, Leiden University Medical Center, Leiden, The Netherlands. [18] Department of Human Genetics, Leiden University Medical Center, Leiden, The Netherlands. [19] Genome Institute of Singapore, A*STAR, Singapore, Singapore. [20] Medical Genetics Department, KOC University, School of Medicine, Istanbul, Turkey. *Lists of authors and their affiliations appear at the end of the paper. ✉email: j.van.dongen@vu.nl

Monozygotic (MZ) twin births arise when the progeny of a single fertilized egg cell, the zygote, divides into two or more embryos early in development. Why and how often this happens is a long-standing enigma of human developmental biology. Nine-banded armadillos produce obligatory MZ quadruplets, but this reproductive strategy is otherwise unique in vertebrate species[1]. Some mammals including humans occasionally produce MZ twins. In humans, MZ twin pregnancies have an increased risk for obstetric, perinatal, and neonatal complications[2–5]. Overall, as many as 12% of human pregnancies may start as multiple pregnancies, but under 2% carry to term[6], resulting in a vanishing twin (the contribution of MZ and dizygotic multiples to these numbers is unknown). MZ twinning rarely runs in families, the prevalence is similar across the world over time (3–4 per 1000 births)[7,8], and stable with the mother's age[7,9,10]. A prevailing hypothesis, therefore, is that MZ twinning occurs at random.

The MZ twinning event occurs early in development, when major epigenetic reprogramming takes place. Starting shortly after fertilization, the methylome of the pre-implantation embryo undergoes multiple waves of global DNA demethylation, followed by de novo methylation[11] as pluripotent cells become committed to different lineages. DNA methylation is essential for embryonic development[12]. In differentiated cells, methylation of CpG islands and shores contributes to the silencing of embryonic genes and imprinted genes, in concert with other repressive epigenetic modifications such as histone marks[12,13].

Here, we show that MZ twins carry a robust DNA methylation signature in somatic tissues. We establish that DNA methylation differences in MZ twins are not randomly distributed across the genome, but are enriched near telomeres and centromeres, at Polycomb-repressed regions and heterochromatin, at genes involved in processes including cell adhesion, WNT signaling, and cell fate, and at putative human metastable epialleles. Finally, we train an epigenetic predictor to recognize MZ twins and show that this tool can predict with reasonable accuracy if an individual is an MZ twin based on a blood or buccal DNA methylation profile, without the need to obtain DNA from both co-twins.

## Results

**Multi-cohort analysis identifies a DNA methylation signature in adult somatic tissues of MZ twins.** To examine if MZ twinning is linked to epigenomic profiles, we analyzed DNA methylation-array data from samples from six independent twin cohorts (Table 1). Dizygotic (DZ) twins constituted the control group. DZ twins represent the ideal control group for detecting a DNA methylation signature of MZ twinning because DZ twins, like MZ twins (but unlike singletons), experience the unique prenatal condition of sharing a womb with a co-twin, thus controlling for possible effects of sharing a womb with a co-twin.

The discovery analysis (total sample size = 1957 individuals) was performed on whole blood DNA methylation data from 924 MZ twin individuals (one randomly selected twin from each MZ pair) and 1033 DZ twin individuals (419 twin pairs; both twins of a pair were included) from the Netherlands Twin Register (NTR). The epigenome-wide association study (EWAS) identified 243 epigenome-wide significant ($p < 1.20 \times 10^{-7}$, Bonferroni correction for 411,169 tests) differentially methylated positions (DMPs) between MZ and DZ twins. Absolute differences in methylation ranged from 0.3 to 6% (0.003–0.06 on the methylation $\beta$-value scale), with a mean of 2.2%. Replication analysis in four independent twin cohorts revealed strong concordance of effects (Fig. 1a–d and Supplementary Data 1): correlations of effect sizes ranged from 0.84 to 0.97. The number of DMPs that replicated following Bonferroni correction for 243 tests ranged from 5 to 186. Since effect sizes were very similar across cohorts (Fig. 1a–d), differences between cohorts in the number of DMPs that replicated following stringent Bonferroni correction likely reflect power related to the following differences between replication cohorts: total sample size (ranging from 356 to 1708), zygosity frequencies (ranging from 33 to 80% MZ), and whether correction for inflation of test statistics was required (Supplementary Table S1). DMPs identified in blood samples from adult twins also showed strong concordance of effects in buccal samples from children (Fig. 1e; $r = 0.87$). Buccal samples consist mainly of epithelial cells[14,15] (predicted mean in the NTR child cohort = 81%), which are derived from the ectodermal cell layer, while white blood cells are derived from the mesodermal lineage. Sensitivity analyses were conducted in the NTR discovery cohort and in the Brisbane Systems Genetics Study (BSGS), because these cohorts also had DNA methylation data available for non-twins (Table 1). These analyses included a comparison of MZ twins to non-twins (parents and siblings) rather than DZ twins, comparison of single MZ twins to single DZ twins (random exclusion of one twin from each pair), and sex-stratified analyses (Supplementary Data 2). The 243 sites showed highly consistent effect sizes across all analyses (Supplementary Note 3, Supplementary Figs. S1, S2, and Supplementary Table S2). By contrast, a comparison of DZ twins to non-twins yielded no epigenome-wide significant DMPs, and showed no strong concordance of effect sizes compared to the main (MZ vs DZ twin) analysis (Supplementary Table S2 and Supplementary Fig. S1), indicating that the results from our primary EWAS (mainly) reflect differential DNA methylation in MZ twins.

We next combined all blood EWAS results (Supplementary Table S3) in a meta-analysis (total sample size = 5723, 88% of samples), which revealed 834 Bonferroni-significant CpGs, hereafter referred to as "MZ-DMPs" (Fig. 2a and Supplementary Data 3): 497 (60%) of which had a lower methylation level in MZ twins (MZ-hypo-DMPs) and 337 had a higher methylation level (MZ-hyper-DMPs). The correlation between MZ-DMPs is described in Supplementary Note 4 and Supplementary Fig. S3.

**Table 1 Cohort descriptives.**

| Cohort | N total EWAS | N MZ twins | N DZ twins | N family members[a] | % female | Age, mean (SD) | Tissue | Array |
|---|---|---|---|---|---|---|---|---|
| NTR | 1957 | 924 | 1033 | 237[a] | 65.3 | 34.9 (11.3) | Blood | 450k |
| E-Risk | 1164 | 470 | 694 | – | 48.9 | 18 (0.4) | Blood | 450k |
| FTC | 1708 | 559 | 1149 | – | 63.7 | 38.6 (20.2) | Blood | 450k/EPIC |
| TwinsUK | 492 | 395 | 97 | – | 100 | 58 (10.1) | Blood | 450k |
| BSGS | 356 | 134 | 222 | 257[a] | 48.9 | 21.4 (14.1) | Blood | 450k |
| NTR | 765 | 564 | 201 | – | 48.8 | 9.6 (1.8) | Buccal | EPIC |

N number of persons (numbers refer to twin individuals, not twin pairs).
[a]Family members of twins (siblings and parents) were not included in the primary EWAS meta-analysis; they were included in sensitivity analyses, and in penalized regression model analyses.
Netherlands Twin Register (NTR), Environmental Risk Longitudinal Twin Study (E-Risk); Finnish Twin Cohort (FTC); UK Adult Twin Registry (TwinsUK); Brisbane Systems Genetic Study (BSGS).

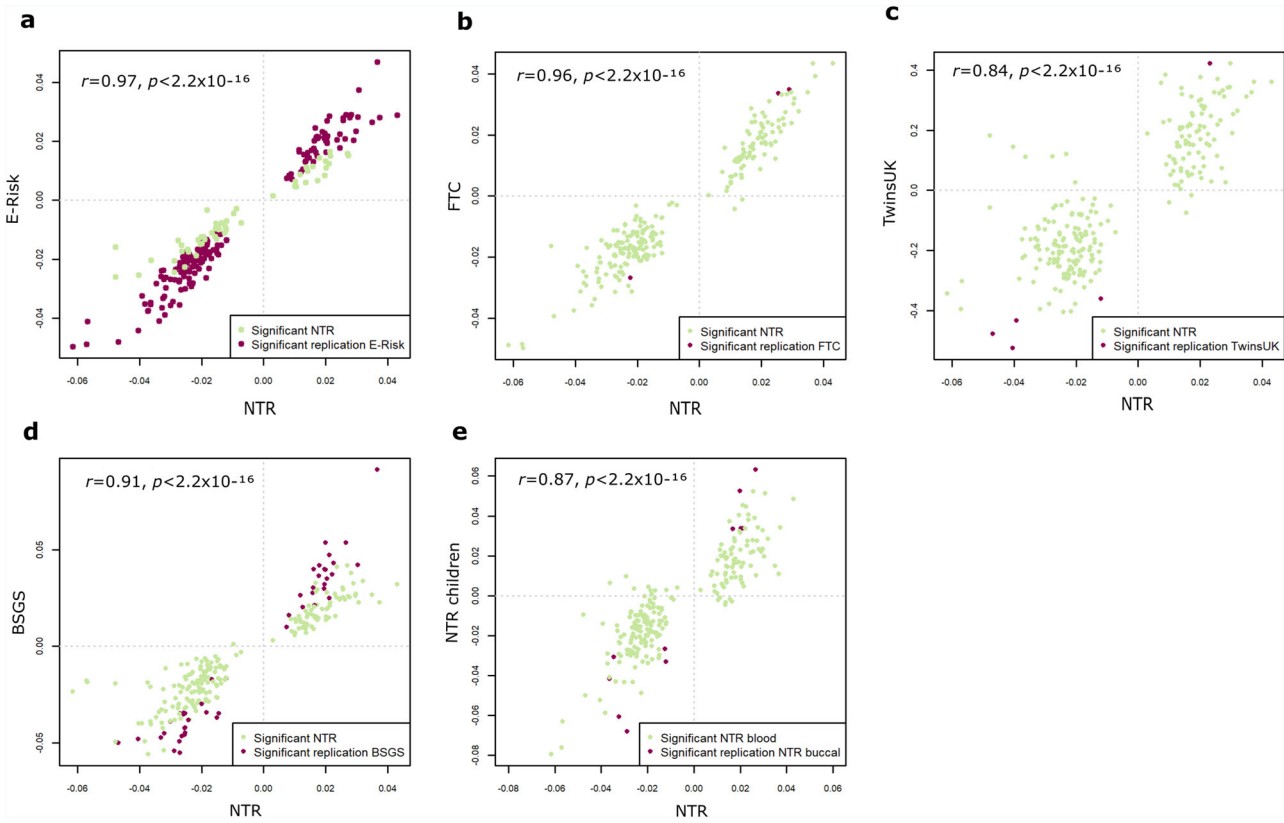

**Fig. 1 Replication of MZ-DMPs identified in NTR in blood DNA methylation data from four independent twin cohorts and buccal DNA methylation data from one independent twin cohort.** Scatterplots showing the estimates (methylation beta-value difference between MZ twins and controls; a positive difference corresponds to a higher methylation level in MZ twins) in the discovery and replication cohorts for MZ-DMPs identified in NTR. The x axis shows the estimates in the discovery EWAS in the Netherlands Twin Register (NTR, $N = 1957$, whole blood). The y-axis shows the estimates in **a** The Environmental Risk Longitudinal Twin Study (E-Risk, $N = 1164$, whole blood). **b** The Finnish Twin Cohort (FTC, $N = 1708$, whole blood). **c** The UK Adult Twin Registry (TwinsUK, $N = 492$, whole blood). In TwinsUK, residuals obtained after correcting for covariates were analyzed instead of methylation beta values. **d** The Brisbane Systems Genetic Study (BSGS, $N = 356$, whole blood). **e** An independent child data set from the NTR ($N = 765$, buccal). Each dot is one methylation site. Methylation sites that replicate following stringent Bonferroni correction for 243 tests and after correction for inflation of genome-wide test statistics, where applicable, are displayed in dark purple, all other sites are shown in green. $r$ = correlation.

**Stability and cross-tissue correlations.** To gain insight into the stability of the 834 MZ-DMPs, we characterized the correlation between longitudinal DNA methylation levels in repeat blood samples collected at an average interval of 5 years. MZ-DMPs showed on average an intermediate methylation level in the blood (mean = 0.52, Supplementary Fig. S4a) and were highly stable over time (mean longitudinal correlation = 0.85, Supplementary Fig. S4b). To examine to what extent methylation variation in the blood reflects variation in other tissues, we examined the correlation between blood samples buccal samples of the same individuals[16], and between blood samples and brain samples[17]. Some CpGs showed strong correlations between blood and buccal cell methylation levels (mean $r = 0.44$) or between blood and brain (mean $r = 0.27$–0.43; for four different brain regions).

**Twin correlations and heritability.** To gain insight into the causes of variation in DNA methylation levels at MZ-DMPs, we characterized total heritability and SNP heritability of DNA methylation in blood[16]. In comparison to genome-wide methylation sites, MZ-DMPs had a high total heritability (mean heritability MZ-DMPs = 57%, mean heritability genome-wide autosomal methylation sites = 19%[16]) and SNP heritability (mean SNP heritability MZ-DMPs = 14%, mean SNP heritability genome-wide autosomal sites = 7%[16]; Supplementary Fig. S5). Most notable is the pattern of twin correlations in the blood

(Supplementary Fig. S6), with MZ twin correlations of the methylation level of the 834 sites being almost three times larger on average compared to DZ twin correlations (mean MZ correlation = 0.58, mean DZ correlation = 0.20). In line with the moderate to large MZ twin correlations at MZ-DMPs, histograms of within-pair differences (Supplementary Fig. S7) and scatterplots (twin 1 versus twin 2) of methylation levels in MZ twin pairs (Supplementary Fig. S8) illustrate that at each MZ-DMP, most MZ pairs have highly concordant methylation levels. Mean absolute within-MZ pair differences in DNA methylation level ranged from 0.8 to 7.8% for different MZ-DMPs (mean = 3.6%). MZ-DMPs with a larger mean difference between MZ and DZ twins also tended to display a larger mean MZ within-pair difference ($r = 0.69$, $p < 2.2 \times 10^{-16}$, Supplementary Fig. S9). Within-pair differences at MZ-DMPs typically showed wide distributions (Supplementary Fig. S10), which illustrates that each CpG displayed more pronounced differences in a subset of MZ pairs. Similarly, each MZ pair showed large within-pair differences at a subset of MZ-DMPs.

We next compared DNA methylation in buccal cells from dichorionic MZ twins (believed to be the earliest splitting type of identical twinning representing 1/3 of all MZ twins), monochorionic diamniotic MZ twins (the most common form of MZ twins representing 2/3 of all cases) and last, monochorionic mono-amniotic MZ twins (who represent less than 1% of MZ twins and are believed to represent a late splitting event)[3]. It has been

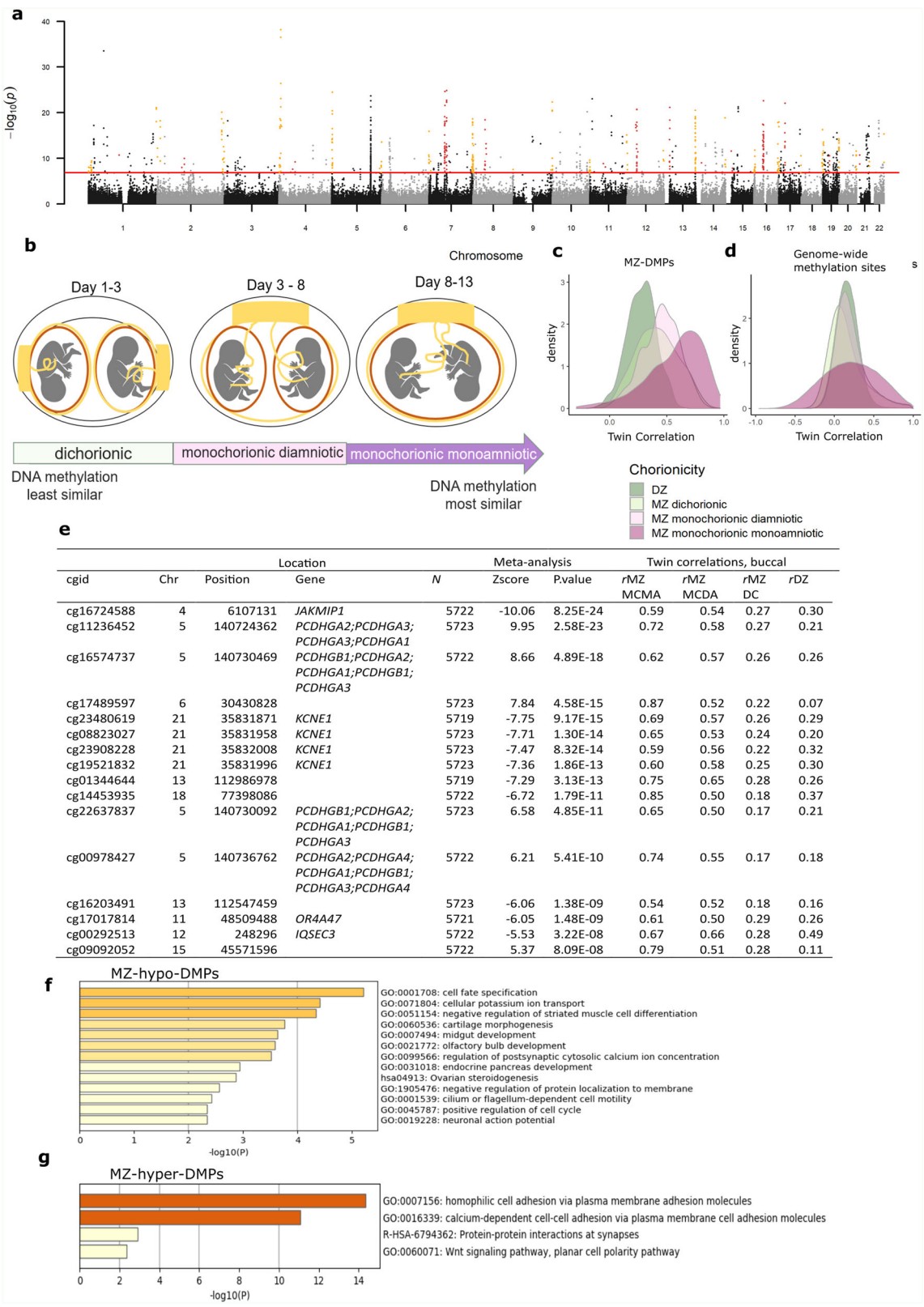

hypothesized that dichorionic MZ twins result from separation soon after fertilization, whereas monochorionic twins result from separation ≥3 days after fertilization[18] (Fig. 2b). Data on chorionicity were obtained by linking data from the NTR child cohort to the Pathological Anatomy National Automatic Archive of the Netherlands (PALGA) database and biobank[19]. Of the 834 MZ-DMPs, 833 were present in the methylation data set from

children. Twin correlations were largest, on average, in MZ monochorionic monoamniotic twins (mean $r = 0.57$, $N = 14$ pairs), followed by monochorionic diamniotic twins (mean $r = 0.45$, $N = 101$ pairs), and MZ dichorionic twins (mean $r = 0.41$, $N = 69$ pairs; Fig. 2c), in contrast to all genome-wide methylation sites, which did not show this differential pattern (Fig. 2d and Supplementary Fig. S11). The distributions of the

**Fig. 2 MZ twinning DMPs identified in a meta-analysis of data from 5723 twins. a** Manhattan plot of the EWAS meta-analysis based on whole blood DNA methylation data from five twin cohorts (total sample size = 5723) that identified 834 MZ-DMPs. The red horizontal line denotes the epigenome-wide significance threshold (Bonferroni correction). Dark red dots highlight significant DMPs near centromeres. Orange dots highlight significant DMPs near telomeres. **b** Dichorionic (DC) MZ twins have separate chorions, amnions, and placentas. Monochorionic diamniotic (MCDA) MZ twins have separate amnions and a common chorion and placenta. Monochorionic monoamniotic (MCMA) have a common chorion, amnion, and placenta. It has been hypothesized that DC MZ twins result from separation soon after fertilization, whereas MC twins are thought to result from separation ≥3 days after fertilization, with MCMA twins arising later than MCDA twins. **c** Density plots of twin correlations for the differentially methylated positions in monozygotic twins (MZ-DMPs) identified in the EWAS meta-analysis illustrate that the overall distribution of twin correlations at MZ-DMPs show the following pattern: rMZ-MCMA > rMZ-MCDA > rMZ-DC. **d** Twin correlations for genome-wide autosomal methylation sites do not follow this pattern. MZ monozygotic twins, DZ dizygotic twins. **e** MZ-DMPs with larger correlations in monochorionic MZ twins compared to dichorionic MZ twins. The CpGs were selected by three criteria (1) rMZ-MCMA > rMZ-MCDA > rMZ-DC; (2) rMZ-MCDA > 0.5; (3) rMZ-DC < 0.2. cgid = Illumina CpG identifier. Chr = chromosome; rMZ-MCMA = correlation in monozygotic monochorionic monoamniotic pairs. rMZ-MCDA = correlation in monozygotic monochorionic diamniotic pairs. rMZ-DC = correlation in monozygotic dichorionic pairs. **f–g** Pathway enrichment analysis results based on the nearest genes of the 834 Bonferroni-significant MZ-DMPs identified in the meta-analysis. **f** Top enriched gene ontology (GO) pathways for MZ-hypo-DMPs (differentially methylated positions with a lower methylation level in monozygotic twins). The darker the color, the stronger the enrichment. **g** Top enriched gene ontology (GO) pathways for MZ-hyper-DMPs (differentially methylated positions with a higher methylation level in monozygotic twins). The darker the color, the stronger the enrichment.

correlations illustrate that the relatively small mean differences are driven by a subset of CpGs that have a larger correlation in monochorionic (especially monochorionic monoamniotic) MZ twins (Fig. 2c). Among CpGs with the most distinctive pattern of monochorionic versus dichorionic twin correlations (Fig. 2e) are multiple CpGs in the protocadherin (*PCDH*) superfamily gene clusters on chromosome 5q31 and multiple CpGs in *KCNE1*; a potassium channel involved in cardiac QT interval.

**Differential methylation occurs near telomeres and centromeres, in regions with repressed chromatin states, and at putative metastable epialleles.** Differentially methylated positions (DMPs) between MZ and DZ twins (MZ-DMPs) were not randomly distributed across the genome. To examine whether DMPs were enriched near telomeres and centromeres, we made a classification to indicate if a CpG was located within a distance of telomeres or centromeres equivalent to 5% of the total chromosome length. Hypomethylation was enriched near telomeres (harboring 45% of MZ-hypo-DMPs, Supplementary Table S2), while hypermethylation was enriched near centromeres (harboring 41% of MZ-hyper-DMPs, Supplementary Table S3). Both hypo- and hypermethylated DMPs were enriched in CpG islands and intergenic regions (Supplementary Tables S4 and S5). Testing for overlap with 15 chromatin states in all cell types from the Roadmap Epigenomics project[20] revealed that hypomethylated sites and hypermethylated sites are enriched in distinct chromatin states. MZ-hypo-DMPs were most strongly enriched in Polycomb-repressed regions characterized by H3K27me3 across most cell types (Supplementary Fig. S12), which are typically associated with transcriptionally-silenced developmental genes. MZ-hyper-DMPs showed significant enrichment in two chromatin states: heterochromatin, and "ZNF genes & repeats", both of which are typically associated with transcriptionally-repressed regions (Supplementary Fig. S13). Given the link between MZ twinning and imprinting disorders, we also examined imprinted loci, but none of the MZ-DMPs overlapped with DMRs of established imprinted loci[21] (Supplementary Fig. S14). Finally, we examined putative human metastable epialleles (MEs), which were previously described to exhibit "epigenetic supersimilarity" between MZ co-twins[22]. MEs are loci with systemic (cross-tissue) inter-individual variation in DNA methylation[23]. Human putative MEs are enriched near-certain classes of transposable elements and exhibit intermediate methylation states. These states can be influenced by genotype, periconceptional environmental exposures, and presumably stochastic processes. Previously defined putative MEs[22] were significantly enriched among the 834

MZ-DMPs (Supplementary Fig. S15), especially among hypo-DMPs (hypomethylated: 11% are putative MEs, $X^2 = 1191.9$, $df = 1$, $p < 2.2 \times 10^{-16}$; hypermethylated: 6% are putative MEs, $X^2 = 233.4$, df = 1, $p < 2.2 \times 10^{-16}$).

**Transcription factor motif and pathway analyses point to early genome programming, early embryonic development, and cell adhesion.** To examine potential functional consequences of MZ-DMPs, we first assessed whether MZ-DMPs overlap with transcription factor binding (TF) sites. In TF motif analysis MZ-hypo-DMPs were significantly enriched within 31 TF motifs (Supplementary Fig. S16 and Supplementary Data 4) including Distal-Less Homeobox 1 (*DLX1*), Engrailed Homeobox 1 (*EN1*), *EN2*, Estrogen Related Receptor Alpha (*ESRRA*), ESX Homeobox 1 (*ESX1*), Gastrulation Brain Homeobox 2 (*GBX2*), Orthodenticle Homeobox 1 (*OTX1*) and SRY-Box Transcription Factor 10 (*SOX10*). For MZ-hypo-DMP-associated TF motifs, gene-based pathway analysis revealed significant enrichment for biological processes involved in "anterior/posterior pattern specification" (Bonferroni-corrected $p = 3.3 \times 10^{-2}$), "chordate embryonic development" (Bonferroni-corrected $p = 4.4 \times 10^{-2}$) and "tube development" (Bonferroni-corrected $p = 2.1 \times 10^{-2}$, for more results see Supplementary Data 5). Pathway analyses based on the nearest genes of DMPs indicated that hypo-DMPs were most strongly enriched for "cell fate specification" (Fig. 2f and Supplementary Data 6), which is driven by several early-expressed transcription factors (Supplementary Fig. S17).

MZ-hyper-DMPs were significantly enriched within 13 TF motifs including motifs for MYC Associated Factor X (*MAX*), Retinoid X Receptor Alpha (*RXRA*), Retinoic Acid Receptor Gamma (*RARG*), Sterol Regulatory Element Binding Transcription Factor 2 (*SREBF2*), Interferon Regulatory Factor 3 (*IRF3*), Nuclear Receptor Subfamily 1 Group H Member 3 (*LXR*), Zinc Finger Protein 691 (*ZFP691*) and Forkhead Box O6 (*FOXO6*) (Supplementary Fig. S18 and Supplementary Data 7). For MZ-hyper-DMP-associated TF motifs, significant enrichment for biological processes involved in retinoic acid receptor signaling was found in gene-based pathway analysis (Bonferroni-corrected $p = 1.3 \times 10^{-3}$, among other developmental signaling pathways (Supplementary Data 8). Pathway analyses based on the nearest genes of DMPs indicated that MZ-hyper-DMPs were most strongly enriched for genes involved in cell-adhesion pathways (Fig. 2g and Supplementary Data 9). Weaker enrichment was seen for the WNT, planar cell polarity (PCP) signaling pathway (GO: 0060071). The enrichment of cell-adhesion pathways is mainly driven by the *PCDH* superfamily gene clusters on

chromosome 5q31 (Supplementary Fig. S19), which showed hypermethylation at 79 CpGs in MZ twins across 614 kb containing the clustered alpha, beta, and gamma protocadherins (Supplementary Fig. S3b).

**Ageing**. Several of the regions that show enrichment among MZ-DMPs, including sub-telomeric regions[24], Polycomb-repressed regions[25], and the protocadherin gene clusters[26] have also been associated with longevity or ageing. We, therefore, looked in more detail at the overlap of age-related DNA methylation variation and MZ-DMPs. We tested for enrichment of methylation sites previously associated with >400 traits reported in the EWAS atlas[27] (Supplementary Note 5 and Supplementary Tables S6, S7) including age-DMPs (i.e. CpGs whose mean methylation level correlates with age) from a number of studies. In addition, we tested for enrichment of age-VMPs (i.e. CpGs whose variance correlates positively with age)[26]. While age-DMPs were not enriched among MZ-DMPs (Supplementary Tables S6 and S7), age-VMPs were significantly enriched among both MZ-hypo and MZ-hyper-DMPs (Supplementary Tables S8 and S9).

**Methylation QTL analyses**. We obtained methylation QTL (mQTL) results for the 834 DMPs (497 hypomethylated, and 337 hypermethylated; Supplementary Data 10 and 11) from our EWAS in the largest mQTL catalog to date; the whole blood mQTL results from the Genetics of DNA Methylation Consortium (GoDMC, $N = 27,750$)[28]. This revealed 108,241 significant *cis* associations between 365 hypo-DMPs and 61,823 genetic variants and 77,988 significant *cis* associations between 196 hyper-DMPs and 35,899 variants. In addition, there were 8197 significant *trans* associations between 4166 variants and 73 hypo-DMPs and 2890 significant *trans* associations between 2116 variants and 52 hyper-DMPs. *Trans* mQTLs were associated with up to 15 CpGs (average = 1.8). Among the genes annotated to *trans* mQTLs were key epigenetic modifiers including *TRIM28* (*trans* mQTL for hypomethylated DMPs) and the de novo methyltransferase *DNMT3B* (*trans* mQTL for hypomethylated DMPs), and a large number of zinc finger genes (for both hypomethylated and hypermethylated DMPs). SNPs with the largest number of *trans* effects were annotated to the *ZNF* gene cluster on chromosome 19 (up to 15 CpGs), and *DPPA4*[29,30], which encodes a key regulator of developmental pluripotency that interacts with the Polycomb Repressor Complex[31] (SNPs rs1044266, rs1163441, and rs2930074, each associated with 11 CpGs in *trans*). Dppa4 forms a heterodimer with Dppa2[32]. In line with the enrichment of hypomethylated DMPs within Polycomb-repressed chromatin states, *DPPA2* and *DPPA4* are *trans* mQTLs for hypomethylated DMPs. In conclusion, methylation Quantitative Trait Locus (mQTL) data[28] provided further support for early-life epigenetic programming, by showing that *trans* mQTLs of MZ-DMPs are annotated to key epigenetic modifier loci such as *TRIM28* and *DNMT3B*, and key pluripotency regulators *DPPA4*[29,30] and *DPPA2*[32]. Further information on mQTL analyses is described in Supplementary Note 6 and Supplementary Fig. S20.

**Retrospective diagnosis of MZ twinning using a DNA methylation-based classifier**. The robustness and enduring properties of DNA methylation in MZ twins suggest that a DNA methylation-based score for MZ twinning could open up new avenues to investigate the link between MZ twinning and congenital disorders with a higher rate of MZ twins among affected individuals[33–38], such as Beckwith–Wiedemann Syndrome (MIM130650), with an almost 10-fold higher frequency of MZ twins[39,40]. For such disorders, it has been hypothesized that

affected singletons began life as a pair of MZ twins in the womb, without the mother's knowledge (vanishing twin syndrome[41,42]). At present, tools to investigate such hypotheses are completely lacking. To examine whether an epigenetic biomarker can be constructed that indicates if a person is an MZ twin, we trained a DNA methylation-based classifier of MZ twinning with penalized regression models (elastic net) on blood DNA methylation data from MZ twins, DZ twins, and family members from NTR ($N \sim$ 2000, Supplementary Note 7 and Supplementary Data 12). We compared models based on two input sets (genome-wide methylation sites versus meta-analysis DMPs), and trained on two phenotypes (MZ versus DZ twins, and MZ twins versus everyone else (including DZ twins and family members of twins)). Predictors trained on the smaller input set of meta-analysis DMPs performed better compared to predictors trained on genome-wide sites, but whether we trained on MZ versus DZ twins or on MZ versus everyone else had little impact on the performance. The area under the curve (AUC) of the best-performing predictors were 0.77 and 0.80, respectively, in an independent blood data set from NTR ($N \sim 1000$) and in blood data from a second independent twin cohort ($N = 606$, BSGS). The predictor performed similarly on methylation data from buccal ($N = 1237$) and on methylation data from monochorionic or dichorionic MZ twins.

## Discussion

These findings show that monozygotic twinning is associated with a persistent DNA methylation profile in adult somatic tissues. This MZ-signature comprises 834 CpG sites enriched in Polycomb-repressed regions and heterochromatin, genes involved in cell adhesion, WNT signaling, cell fate, and putative MEs. MZ-DMPs were strongly enriched near telomeres and centromeres. We anticipate these findings to be the starting point for further functional studies aiming to reconstruct the precise molecular events leading to division of the zygote. Despite the link between MZ twinning and imprinting disorders, none of the MZ-DMPs map to known imprinted genes.

We have identified a strong epigenetic signature associated with MZ twinning, a phenomenon that typically shows little evidence for a genetic component. Nevertheless, we found support for MZ-DMPs being influenced by common variants, and MZ twin correlations of the methylation level of the 834 sites were on average almost three times larger compared to DZ twin correlations This pattern is consistent with strong genetic influences on DNA methylation, with allelic or gene-gene interactions. Of note, rare cases of familial MZ twinning with an autosomal dominant inheritance have been described[9,43,44]. An alternative explanation for the pattern of twin correlations could be that the methylation state is established in the early zygote prior to the separation that leads to MZ twins, and subsequently inherited across mitosis[22]. In this case, DNA methylation level will be more similar in MZ twins because they are derived from one zygote, compared to DZ twins who are derived from two zygotes. Our sensitivity analyses showed that MZ-DMPs were unaffected by correction for nearby common variants (*cis* mQTLs). Future studies with genome sequencing and methylation data from MZ twins may establish if the MZ-signature is linked to rare sequence variants carried by MZ twins. A recent study reported that post-zygotic de novo mutations, including presumed post-twinning CpG > TpG mutations (which may affect DNA methylation), are not uncommon in MZ twins, but did not report if these mutations occur more frequently in MZ twins than in other individuals[45].

Several clues emerge from our pathway analyses. First, the (proto)cadherin gene signal raises the possibility that cell

adhesion might be involved in the MZ twinning process. Cell adhesion could be associated with the tendency of an embryo to dissociate, perhaps during early cleavage stages. Although the enrichment of cell adhesion was mainly driven by the *PCDH* superfamily gene clusters on chromosome 5q31, one cadherin gene *CELSR3*; a member of the WNT/PCP signaling pathway, also showed hypermethylation in MZ twins. Noteworthy, cadherins have been proposed in early candidate gene studies of MZ twinning[46]. Second, MZ-DMPs may occur at genes and within binding sites of TFs involved in early embryonic development. We found further support for early-life epigenetic programming through mQTL analyses.

Previously established MEs[22] were strongly enriched among MZ twinning DMPs. The enrichment of MEs among MZ-DMPs is particularly interesting because their methylation state is established around implantation of the embryo[47], when the majority of MZ twinning events are thought to occur (producing monochorionic diamniotic MZ twins). The methylation level at MZ-DMPs was on average more strongly correlated between monochorionic MZ twins compared to dichorionic MZ twins. Importantly, the exact timing of MZ twinning is unknown to date and the connection between the time of splitting and chorionicity remains a hypothesis. A limitation of our chorionicity analyses is that the sample size was relatively small due to the limited numbers of twins for which reliable chronicity data and DNA methylation data were available.

MZ-DMPs were strongly enriched near telomeres and centromeres, but the reason for this remains to be established. We note that sub-telomeric regions[24], as well as several other regions that show enrichment among MZ-DMPs, namely, Polycomb-repressed regions[25], and the protocadherin gene clusters[26] have been previously associated with longevity or ageing. Furthermore, we observed that age-variable methylated positions, i.e. CpGs whose variance correlates positively with age, were significantly enriched among MZ-DMPs. We interpret the overlap with ageing-related loci as an indication that MZ twinning and age-related epigenetic dysregulation both affect loci whose methylation pattern is established early in development. We note that the observation is unlikely to have implications for the lifespan of MZ twins, because data from the largest population-based and oldest twin cohort ever studied (109,303 twins from the Danish Twin Registry born between 1870 and 1990) concluded that monozygotic and dizygotic twins have similar lifespans[48].

A question that follows from our findings is whether there might be implications for epigenetic studies in twins in general. To consider this question, it is important to note that the 834 MZ differentially methylated sites represent only a tiny fraction of all genome-wide sites, although it is unknown whether more differentially loci might exist that may be uncovered by future larger EWA studies, or by other techniques with greater coverage (e.g. bisulfite sequencing). Second, whether any of the methylation differences have phenotypic consequences for MZ twins is unknown. We do not foresee these findings to impact the generalizability of (discordant) MZ twin studies that aim to detect epigenetic variation connected to a trait or disease, unless the trait is connected to the MZ twinning process itself. The literature suggests that for most outcomes studied, MZ twins are very comparable to non-twins, except for some congenital disorders, birth weight, and traits strongly related to birth weight[49]. MZ-DMPs showed MZ twin correlations that were on average more than three times as large as the DZ twin correlations, which can indicate a strong genetic influence (i.e. DNA sequence effect) on these DNA methylation sites or be indicative of mitotic inheritance of a pre-twinning established methylation state. The latter scenario has implications for the interpretation of classical twin studies of epigenetic marks in which correlation patterns of MZ and DZ twins are contrasted to estimate the heritability of epigenetic marks (but as noted, this remark only applies to a limited set of loci in the genome).

We reported some loci with methylation differences extending across large regions of correlated CpGs (such as the *PCDH* superfamily gene clusters), but note that this finding should be interpreted with some caution, because correlations between methylation levels at different CpGs may also arise due to underlying DNA sequence similarity. We excluded probes affected by SNPs and cross-reactive probes (probes that are predicted to hybridize to multiple locations in the genome based on large sequence similarity) from all analyses, as is common practice in EWA studies, and flagged a small number of MZ-DMPs that are measured by probes with a smaller degree of sequence similarity to multiple regions. We note that cross-hybridization arising from even a very small degree of probe off-target similarity (i.e. 14 bases) can affect EWAS results in certain cases, namely when the phenotype studied is associated with a repeat expansion[50]. The extent to which EWA studies in general, including ours, are affected by such low levels of probe overlap with other genomic sequences is unknown. We note that excluding probes with some off-target sequence is undesirable as it would lead to the removal of practically all probes[50]. Rather, future studies with sequencing-based techniques could provide insight into the methylation signature of MZ twins at full resolution, and address the potential contribution of underlying DNA sequence variation.

A connection of MZ twinning to the early establishment of DNA methylation (prior to the formation of distinct cell lineages) could explain the strong replication of our results across different cohorts and different tissues. It is well-established that very early-life exposures such as periconceptional maternal nutrition[51,52], cellular differentiation[20], and genetic variation[28], are associated with persistent DNA methylation changes in adult somatic tissues of progeny. MZ twinning may be regarded as an unusual early-life event of the embryo. We interpret the MZ twinning DMPs as representing a molecular signature of the MZ twinning event that persists, through many rounds of mitosis, to adult somatic tissues. Whether these methylation differences represent a cause, effect, or byproduct of the MZ twinning event remains to be determined.

## Methods

**Overview**. An epigenome-wide association study (EWAS) with MZ twins as cases and DZ twins as controls was performed on whole blood DNA methylation data (Illumina 450k array) from adult participants of the Netherlands Twin Register[53] (NTR, discovery cohort) to identify differentially methylated positions (DMPs). Replication analyses were carried out in four cohorts (replication cohorts) with whole blood DNA methylation data available in adults: the UK Adult Twin Registry[54] (TwinsUK, Illumina 450k array), the Environmental Risk Longitudinal Twin Study[55] (E-Risk, Illumina 450k array), the Finnish Twin cohort[56] (FTC, Illumina 450k and EPIC array), and the Brisbane Systems Genetics Study[57,58] (BSGS, Illumina 450k array). A cross-tissue replication analysis was carried out in an independent cohort of children from the NTR (NTR-ACTION) with DNA methylation measured in buccal cells with the Illumina EPIC array[59]. Epigenome-wide significance was assessed using Bonferroni correction for the number of sites tested ($0.05/411169$; alpha $= 1.20 \times 10^{-7}$). Replication of top sites from the discovery cohort was evaluated following Bonferroni correction for the number of top sites identified in the discovery cohort ($0.05/243 = 2.1 \times 10^{-4}$), by computing the Pearson correlation between the effect sizes in NTR and the discovery cohort in each replication cohort, and by evaluating the number of sites with the same direction of effect in the discovery cohort and each replication cohort. Next, results from all cohorts were combined in a meta-analysis. Results from the meta-analysis were used as input for follow-up analyses, including analyses of heritability and chorionicity, enrichment analyses of genomic features (i.e. location with respect to CpG islands and gene elements), functional elements (chromatin states), transcription factor binding sites, results from previous EWA studies of traits and exposures, and mQTL analysis.

### Subjects and samples

*Netherlands Twin Register (NTR)*. The participants take part in longitudinal studies with the Netherlands Twin Register (NTR)[53,60,61] including the NTR biobank

project between 2004 and 2011[62]. The NTR is a longitudinal twin-family study with no other selection criteria than being a multiple or one of their family members. In total, good quality DNA methylation data from whole blood were available for 3089 samples from 3057 NTR participants, including monozygotic and dizygotic twins, parents of twins, siblings of twins, and spouses of twins. All familial relationships, including the zygosity of twins, were confirmed by genotype data[16]. In the primary EWAS, we included one randomly selected MZ twin of each pair and complete dizygotic twin pairs for whom the following data were available: good quality DNA methylation data and data on white blood cell counts, body mass index (BMI), and smoking status, leaving 1957 subjects: 924 MZ twins and 1033 DZ twins. Informed consent were obtained from all participants. The study was approved by the Central Ethics Committee on Research Involving Human Subjects of the VU University Medical Centre, Amsterdam, an Institutional Review Board certified by the U.S. Office of Human Research Protections (IRB number IRB00002991 under Federal-wide Assurance-FWA00017598; IRB/institute codes, NTR 03-180).

*Environmental Risk (E-Risk) Longitudinal Twin Study.* The E-Risk study follows a 1994–95 birth cohort of 2232 British children and has been described in detail previously[55]. Home visits of participants took place at ages 5, 7, 10, 12, and most recently, 18 years (93% participation). The Joint South London and Maudsley and the Institute of Psychiatry Research Ethics Committee approved each phase of the study (997/122). Parents gave informed consent and twins gave assent between 5 and 12 years and then informed consent at age 18.

*Finnish Twin Study (FTC).* The Finnish Twin Study is part of three longitudinal cohorts[56,63], the Older Twin cohort[56], FinnTwin16 (FT16)[64], and FinnTwin12 (FT12)[65]. The Older Twin Cohort is comprised of 13,888 same-sex twin pairs born before 1958, while FT16 and FT12 are longitudinal studies of five consecutive birth cohorts (born in 1975–1979, *n* = 2800 pairs, and 1983–1987, *n* = 2700, respectively) of Finnish monozygotic and dizygotic twins who have completed surveys and interviews beginning in adolescence and into adulthood (FT16 at age 16, 17, 18, 24, 34; FT12 at age 11, 14, 17.5, 24). DNA methylation from whole blood was collected from 2616 twins with complete info on body mass index and smoking status. 2231 samples passed quality control from 1082 monozygotic twins and 1149 dizygotic twins.

The EWAS included one randomly selected MZ twin from each twin pair, or the twin whose sample passed quality control in the case that one twin's sample was not of high quality, and all dizygotic twins. Participants were given information about the study procedures and design in their native language (Finnish or Swedish) and provided informed consent, following the principles of informed consent in the Declaration of Helsinki. All study procedures were approved by the ethics committees of Helsinki University Central Hospital (113/E3/2001, 249/E5/2001, 346/E0/05, 270/13/03/01/2008, and 154/13/03/00/2011).

*TwinsUK.* TwinsUK is a nationwide registry of adult twins from the UK with more than 14,000 volunteers recruited through media campaigns without selecting for particular diseases[54]. The majority of participants are adult females of European descent. The registry started in 1992 recruiting middle-aged female twins and from 1995 the invitation was extended to same-sex twins over 18 years old. The current study included 246 female TwinsUK MZ and DZ twin pairs with existing whole blood DNA methylation profiles. Ethical approval was granted by the National Research Ethics Service London-Westminster, the St Thomas' Hospital Research Ethics Committee (EC04/015 and 07/H0802/84). All research participants have signed informed consent prior to taking part in any research activities.

*Brisbane Systems Genetics Study (BSGS).* DNA methylation data were available for 614 individuals from 117 families (European ancestry), including adolescent MZ and DZ twins, their siblings, and their parents, as described in McRae et al. [57]. Participants were selected from the Brisbane Systems Genetics Study[58]. The primary EWAS was performed on data from MZ and DZ twins. Parents and siblings of twins were included in sensitivity analyses and epigenetic predictor analyses. This study was approved by the Human Research Ethics Committee of the QIMR Berghofer Medical Research Institute (Approval number P1176 "Mapping eQTL to dissect the genetic basis of complex trait variation", NHMRC #EC00278). All participants gave informed written consent. DNA was extracted from peripheral blood lymphocytes by the salt precipitation method from samples that were time matched to sample collection of PAXgene tubes for gene expression studies in the Brisbane Systems Genetics Study.

*Netherlands Twin Register (NTR) – ACTION Cohort.* Participants are twin children who take part in longitudinal studies from the Netherlands Twin Register (NTR) and participated in the ACTION project[66–68]. From the population-based NTR, the ACTION study identified twins who at least once scored higher or lower on a sum score for aggression[66]. After quality control, EPIC array methylation was available for 1237 buccal samples from 1235 twins, including 1036 samples from MZ twins and 201 samples from DZ twins. The zygosity of twins was confirmed by genotype data[59]. In the EWAS, we included one randomly selected MZ twin of each pair and complete dizygotic twin pairs with good quality DNA methylation data, leaving 765 subjects (564 MZ twins and 201 DZ twins).

Parents of twins could indicate if they wished to be informed of the results of zygosity testing based on a set of SNPs and VNTRs, as described previously[69]. Informed consent was obtained from parents. The study was approved by the Central Ethics Committee on Research Involving Human Subjects of the VU University Medical Centre, Amsterdam, an Institutional Review Board certified by the U.S. Office of Human Research Protections (IRB number IRB00002991 under federal-wide Assurance-FWA00017598; IRB/institute codes, NTR 03-180).

**Methylation measurements.** DNA methylation was assessed with Illumina BeadChips according to the manufacturer's protocol: the Illumina Infinium HumanMethylation450 BeadChip (450k array), which measures more than 450,000 methylation sites (majority of cohorts), or the Illumina MethylationEPIC BeadChip (EPIC array), which measures more than 850,000 methylation sites. DNA methylation $\beta$-values were analyzed, which range from 0 to 1, indicating the proportion of DNA that is methylated at a specific CpG in a sample. Cohort-specific details about DNA methylation profiling, quality control, and normalization are described below.

*Netherlands Twin Register (NTR).* Blood sampling procedures have been described in detail[62] DNA methylation was assessed with the Infinium HumanMethylation450 BeadChip Kit (Illumina, San Diego, CA, USA) by the Human Genotyping facility (HuGeF) of ErasmusMC, the Netherlands (http://www.glimdna.org/) as part of the Biobank-based Integrative Omics Study (BIOS) consortium[70]. DNA methylation measurements have been described previously[16]. Genomic DNA (500 ng) from whole blood was bisulfite-treated using the Zymo EZ DNA Methylation kit (Zymo Research Corp, Irvine, CA, USA), and 4 μl of bisulfite-converted DNA was measured on the Illumina 450k array following the manufacturer's protocol. A number of sample- and probe-level quality checks and sample identity checks were performed. Quality control and normalization have been described in detail previously[16]. In short, sample-level QC was performed using MethylAid[71]. Probes were set to missing in a sample if they had an intensity value of exactly zero, or a detection p > 0.01, or a bead count of <3. After these steps, probes that failed based on the above criteria in >5% of the samples were excluded from all samples (only probes with a success rate ≥0.95 were retained). The methylation data were normalized with functional normalization[72].

*Environmental Risk (E-Risk) Longitudinal Twin Study.* Blood sampling and methylation measurement procedures have been described in detail previously[73]. Briefly, whole blood was collected at age 18 from 82% (*N* = 1700) of the participants and 1669 blood DNA samples were assayed (31 samples were not useable due to e.g. low DNA concentration). ~500 ng of DNA from each sample was treated with sodium bisulfite using the EZ-96 DNA Methylation kit (Zymo Research, CA, USA). DNA methylation was measured with the Illumina Infinium Human-Methylation450 (Illumina, San Diego, CA, USA). Quality control and normalization have also been described in detail previously[73]. In brief, sample-level QC was performed using the methylumIDAT function in methylumi[74,75]. First, samples with median methylated and unmethylated intensities <2500 were excluded. Second, the efficiency of sodium bisulfite conversion was checked based on ten control probes and samples with a "conversion score" <80 were excluded. Third, multidimensional scaling on sex chromosome DNA methylation probes was compared to reported gender. Fourth, genetic identity was confirmed by comparing 450K array array SNP probe genotype to genotype data based on Illumina OmniExpress24v1.1 genotyping BeadChips. Based on the pfilter function from the wateRmelon R package[76] we excluded: 0 samples with >1% of sites with a detection *p*-value >0.05, 567 sites with bead count <3 in 5% of samples and 1448 probes with >1% of samples with detection *p*-value >0.05. Normalization was performed with the dasen function from the wateRmelon package. Prior to the analyses, probes with SNPs (MAF > 5%) within 10 bp of the single base extension and ambiguous probes were excluded[77,78], resulting in a final data set of 430,802 probes. Samples from 1658 E-Risk twins passed our QC pipeline, including 734 complete twin pairs (58% MZ).

*Finnish Twin Study (FTC).* Blood samples were collected from subjects as part of targeted studies on the twins, including some longitudinal sampling[56,63]. DNA was extracted from whole blood using the QIAamp DNA Mini kit (QIAGEN Nordic, Sollentuna, Sweden) and bisulfite conversion was performed using the EZ-96 DNA MethylationGold Kit (Zymo Research Corp, Irvine, CA, USA) following manufacturer instructions. DNA methylation was measured using the Infinium HumanMethylation450 BeadChip Kit (Illumina, San Diego, CA, USA) for 1412 samples at the Norwegian Genomics Consortium (Norway), The University of Chicago Genomics Facility (Chicago, USA), and SNP & SEQ Technology Platform (Uppsala, Sweden), and the Infinium HumanMethylationEPIC BeadChip Kit (Illumina, San Diego, CA, USA) for 819 samples at Diagenode (Vienna, Austria) and the FIMM Tech Centre (Helsinki, Finland).

We performed several steps to remove samples and probes that did not meet strict quality standards. Samples with poor quality were identified using the R package *MethylAid* with default thresholds for 450k and EPIC data[71]. Additionally, we removed probes with a detection *p*-value > 0.01, an intensity value of exactly 0, or a bead count <3 in more than 5% of samples. 450k and EPIC samples were normalized using were performed using ssNoob[79], and the data sets combined,

retaining only probes present on both arrays and passing QC. Beta-mixture quantile (BMIQ) normalization was used to adjust beta values for differences due to probe type[80].

*TwinsUK.* DNA for methylation assessment was extracted from whole blood and stored in EDTA tubes. The Infinium HumanMethylation450 BeadChip (Illumina Inc, San Diego, CA) was used to measure DNA methylation levels, as previously described[81]. The Infinium HumanMethylation450 BeadChips were processed using the ENmix package[82] to obtain methylation beta values. Briefly, background correction was performed using the Exponential-Normal mixture distribution (ENmix) method using out-of-band type I probe intensities to model background noise, dye-bias correction was performed using the Regression on Logarithm of Internal Control probes (RELIC) method[83], and probe design bias adjustment was performed implementing the Regression on Correlated Probes (RCP) method[84]. Signals with a high detection *p*-value $> 1 \times 10^{-6}$ and a low number of beads <3 were set to missing. Samples with missing data in >5% of the CpGs were excluded. CpGs with missing data in >5% of the samples were excluded. Samples identified as outliers in terms of the bisulfite intensity, total intensity, or beta-value distributions were excluded.

*Brisbane Systems Genetics Study.* Bisulfite conversions were performed in 96 well plates using the EZ-96 DNA Methylation Kit (Zymo Research, Irvine, CA, USA). Prior to conversion, DNA concentrations were determined by NanoDrop quantification (NanoDrop Techologies, Inc., Wilmington, DE, USA) and standardized to include 500 ng. Three technical replicates were included in each conversion to assess repeatability. A commercial female human genomic DNA sample (Promega Corporation, Madison, WI, USA) was used on all plates, one sample from each run was duplicated on the plate and one sample was duplicated from a different plate. DNA recovery after conversion was quantified using Nanodrop (ThermoScientific, Wilmington, DE, USA). Bisulfite-converted DNA samples were hybridized to the 12 sample, Illumina HumanMethylation450 BeadChips using the Infinium HD Methylation protocol and Tecan robotics (Illumina, San Diego, CA, USA). The HM 450 BeadChip-assessed methylation status was interrogated at 485,577 CpG sites across the genome. It provides coverage of 99% of RefSeq genes. Methylation scores for each CpG site are obtained as a ratio of the intensities of fluorescent signals and are represented as *β*-values. Samples were randomly placed with respect to the chip they were measured on and to the position on that chip in order to avoid any confounding with family. Data QC and normalization were conducted using the meffil R package[85]. Default QC threshold parameters were used to exclude samples and DNA methylation sites, followed by functional normalization[4].

*Netherlands Twin Register (NTR) – ACTION Cohort.* The procedures of buccal swab collection have been described previously[62]. DNA methylation was measured with the Infinium MethylationEPIC BeadChip Kit (Illumina, San Diego, CA, USA)[86] by the Human Genotyping facility (HugeF) of ErasmusMC, the Netherlands (http://www.glimdna.org/). DNA extraction, DNA methylation-array measurements, and quality control have been described in detail elsewhere[59]. In brief, the quality control (QC) and normalization of the methylation data were performed using a pipeline developed by the Biobank-based Integrative Omics Study (BIOS) consortium (https://molepi.github.io/DNAmArray_workflow/), which includes sample quality control using the R package MethylAid[71], and probe filtering and functional normalization as implemented in the R package DNAmArray. The R package omicsPrint[87] was used to verify sample relationships-based SNPs (e.g. zygosity of twins). DNAmArray and meffil[85] were used to identify sex mismatches. Data were normalized with functional normalization[72]. The following probe filters were applied: Probes were set to missing (NA) in a sample if they had an intensity value of exactly zero, detection *p*-value > 0.01, or bead count <3. Probes were excluded from all samples if they mapped to multiple locations in the genome, if they overlapped with a Single Nucleotide Polymorphism (SNP) or Insertion/Deletion (INDEL), or if they had a success rate <0.95 across samples. Ambiguous mapping probes (overlap ≥ 47 bases per probe) and probes where genetic variants (SNPs or INDELS) with a minor allele frequency >0.01 in Europeans overlap with the targeted CpG or single base extension site (SBE) were obtained from Pidsley et al[88]. After probe filtering, the success rate of probes for each sample was checked: All samples had a success rate above 0.95 (after removal of low-performing samples detected by MethylAid). Only autosomal methylation sites were analyzed, leaving 787,711 out of 865,859 sites.

## Covariates
*Netherlands Twin Register (NTR).* Measured white blood cell percentages were included as covariates in the EWAS to account for variation in cellular composition between whole blood samples, and were obtained as part of the complete blood count[4]. The following WBC were included as covariates: monocytes, eosinophils, and neutrophils (lymphocyte percentage was not included because it correlated with neutrophils ($r = -0.9$)), and basophil percentage was not included because it showed very little variation between individuals. Body mass index (kg/m²) was computed based on weight and height obtained at the moment of blood sampling. Information on current and past smoking behavior was also collected as part of the NTR biobank project at the moment of blood draw. Smoking status was coded as 0 (never smoked), 1 (former smoker), 2 (current smoker).

*Environmental Risk (E-Risk) Longitudinal Twin Study.* White cell-type proportions were estimated from the methylation data using the Houseman method[89]. Estimated cell types included plasma blasts, $CD8^+CD28^-CD45RA^-$ T cells, naive CD8 T cells, CD4 T cells, natural killer cells, monocytes, granulocytes. Body mass index (kg/m²) was computed based on weight and height obtained at the time of blood sampling. Information on current smoking behavior was also collected at the time of blood sampling. Smoking status was coded as 0 (never smoked) and 1 (current smoker). To permit control for technical variation, we used methylation-array control-probe principal components (PCs)[90]. 28 PCs were needed to explain 90% of the variance.

*Finnish Twin Study (FTC).* White blood cell percentages were estimated using the updated IDOL libraries for 450k and EPIC data[91]. We included the estimated proportions of CD8 T cells, CD4 T cells, natural killer cells, and neutrophils, and excluded monocytes and B cells to prevent multicollinearity. Additionally, we included information on height and weight were used to compute BMI (kg/m²) at the time of blood sampling, and smoking history (never, former, or current smoker) was determined via questionnaire.

*TwinsUK.* Blood cell composition was estimated in minfi[92] using the Houseman algorithm[93], and the proportion of monocytes, eosinophils, and neutrophils were included as covariates in the analyses. Body mass index (kg/m²) was computed based on weight and height obtained at the time of blood sampling. Information on current and past smoking was collected from questionnaires. Smoking status was coded as 0 (never smoked), 1 (former smoker), 2 (current smoker).

*Brisbane Systems Genetics Study.* The following imputed white blood cell percentages were included as covariates in the EWAS of MZ versus DZ twins: monocytes, eosinophils, and neutrophils. Body mass index (kg/m²) was calculated from height in cm and body mass/weight in kg, either (a) in the case of parents, from self-report during the clinical visit (up to 2011) or an online mothers' questionnaire (from 2012); or (b) for twins and their siblings, measured by staff during the clinical visit corresponding to the collection time for the DNA sample. A smoking score was calculated by applying the CpG weights in the DNAm "smoking" predictor described by McCartney et al. [94]. This score was included as a covariate in the analysis comparing MZ twins versus everyone else.

*Netherlands Twin Register (NTR) – ACTION cohort.* Cellular proportions were included as covariates in the EWAS to account for variation in cellular composition between buccal samples. Cellular proportions were predicted with Hierarchical Epigenetic Dissection of Intra-Sample-Heterogeneity (HepiDISH) with the RPC method (reduced partial correlation), as described by Zheng et al[95] and implemented in the R package EpiDISH. Predicted percentages of epithelial and natural killer cells were included as covariates in the EWAS. Other leukocytes were not included in the model because they either had very low levels or correlated strongly ($|r| \geq 0.9$) with other cell counts included in the model (epithelial cells and/or natural killer cells).

## Genotype data
Genotype data were used in sensitivity analyses that were performed in the NTR cohort in which the association between DNA methylation and zygosity was tested while adjusting for the top *cis* mQTL of each methylation site (identified by the GoDMC consortium)[28]. The genotype data used in this analysis have been described previously[28]. In brief, genotyping was done on multiple platforms, with a number of overlapping participants. The following platforms were chronologically used: Affymetrix-Perlegen, Illumina 660, Illumina Omni Express 1M, and Affymetrix 6.0. Genotype calls were made with the platform-specific software (Birdseed, APT-Genotyper, Beadstudio) following manufacturers' protocols. For the Affymetrix-Perlegen and Illumina 660 platforms, the SNPs were lifted over to build 37 (HG19) of the Human reference genome. Per platform, a sample was removed if the call rate for this person was <90%, the Plink 1.07 inbreeding value F was <−0.075 or >0.075, the gender of the person did not match the DNA of the person, the IBD status did not match the expected familial relations, or the sample had more than mean + 5sd Mendelian errors. For the Affymetrix 6.0 platform also all samples with a CQC value < 0.40 were removed. Afterward, in case a subject, was genotyped on multiple platforms, only the platform with the highest number of SNPs was selected when concordance between platforms was over 97%. Allele - and strand alignment of SNPs was done against the Dutch GONL reference panel for each platform[96]. SNPs were removed in each platform when MAF < 0.005, HWE < 10–12 and the call rate of the SNP was <95%[97]. Then SNPs were only selected if the allele frequency of the SNP deviated <0.10 as compared to the GONL data. Subsequently, the individual platform data were merged into a single data set. In this single data set, the sample IBD, on a common backbone of ~70 K SNPs, was re-compared with their expected familial relations and samples were removed if they did not match. The single merged data set was imputed with mach-admix, using GONL as a reference panel, for only the SNPs that survived QC and were present on at least one platform, forcing missing genotype imputation for all SNPs. Best guess genotypes were generated from these data and from these cross-platform imputed SNPs, the following SNPs were selected: SNPs with a $R^2 > 0.90$, with HWE $p > 0.00001$, with a Mendelian error rate

<2% and if the association of one platform = case vs. the other platforms = controls p-value > 0.00001 (of course applied for each platform). This left 1.2 M SNPs. These SNPs were then re-aligned against the 1000 Genomes Phase 3v5 reference and then imputed to that reference on the Michigan imputation server[98]. From the resulting VCF files, best guess genotypes were calculated.

**Epigenome-wide association analysis.** EWAS analyses were performed in R[99]. The difference in methylation level between MZ and DZ twins was tested in generalized estimation equation (GEE) models with DNA methylation $\beta$-value as outcome and the following predictors: zygosity (DZ = 0, MZ = 1), sex (not included in TwinsUK, where all twins were female), age at DNA sampling (not included in E-Risk, where all twins were 18 years old), cellular composition (measured or imputed cell percentages), technical covariates, and in the adults twin cohorts, we also adjusted for BMI and smoking (because BMI and smoking are known to have large effects on methylation)[100,101]. In TwinsUK, instead of the methylation $\beta$-value, the residuals derived after adjusting methylation $\beta$-values for batch effects were analyzed as this was the optimal analysis strategy for this cohort. First, using a linear mixed model DNA methylation $\beta$-values were regressed on technical covariates (array and position in the array) as random effects. The residuals were then used as the outcome variable in a generalized estimation equation (GEE) model, which was fitted with the R package "gee". In the primary EWAS of BSGS (adolescent twins), smoking was not included as covariate. The following probes were removed from all cohorts: sex chromosomes, probes with a single nucleotide polymorphism (SNP) within the CpG site (at the C or G position) irrespective of minor allele frequency in the Genome of the Netherlands (GoNL) population, irrespective of minor allele frequency[70], and ambiguous mapping probes reported by Chen et al. with an overlap of at least 47 bases per probe[78]. In addition, we examined and flagged potentially cross-hybridizing probes at lower levels of sequence overlap (Supplementary Note 4), and examined the sequence similarity of probes underlying significant MZ-DMPs detected in our meta-analysis with the R package DNAmCrosshyb (https://github.com/pjhop/dnamarray_crossreactivity[50]). We considered two previously described stringent thresholds of overlap: ≥30 bases & ≥14 bases[50]. The R package Bacon was used to compute the Bayesian inflation factor and to obtain bias- and inflation-corrected test statistics prior to meta-analysis[102].

All cohorts performed the EWAS analyses with generalized estimation equation (GEE) models, which were fitted with the R package "gee". The following settings were used: Gaussian link function (for continuous data), 100 iterations, and the "exchangeable" option to account for the correlation structure within families. In all cohorts, except for BSGS (which included complete MZ pairs), the EWAS was performed with one, randomly selected MZ twin of each pair removed from the analysis. This conservative approach was applied to rule out any risk that the correction for familial resemblance for this peculiar phenotype would not work adequately; however, sensitivity analyses showed that the results were robust to the inclusion of either single twins or complete twin pairs (Supplementary Note 3).

The exact list of predictors included in the primary EWAS in gee in each cohort was as follows: NTR; zygosity, sex, age at blood sampling, percentages of monocytes, eosinophils, and neutrophils, HM450k array row, 96-wells bisulfite sample plate (dummy coding), smoking status and BMI. E-Risk; zygosity, sex, cell-type estimates (described above), technical PCs (described above), smoking status, and BMI. FTC; zygosity, sex, age at blood sampling, estimated proportions of CD8 T cells, CD4 T cells, natural killer cells, and neutrophils, smoking status, BMI, and array type. TwinsUK; zygosity, age at blood sampling, percentages of monocytes, eosinophils, and neutrophils, smoking status, and BMI. BSGS; zygosity, sex, age at blood sampling, percentages of monocytes, eosinophils, and neutrophils, and BMI. NTR-ACTION cohort: zygosity, sex, age at buccal sample collection, percentages of epithelial cells and natural killer cells, EPIC array row, and 96-wells bisulfite sample plate (dummy coding).

**Sensitivity analyses.** Sensitivity analyses were performed to examine the robustness of the findings. Sensitivity analyses were carried out in the largest sample (NTR). First, we compared the primary EWAS approach with MZ twins as cases (MZ = 1) and DZ twins as controls (DZ = 0 to 1) an EWAS with MZ twins as cases (MZ = 1), and family members (parents and siblings) as controls (parents and siblings = 0, DZ twins were excluded) and (2) and EWAS comparing DZ twins (DZ = 0) to family members (parents and siblings) as controls (parents and siblings = 1, MZ twins were excluded). Second, we compared the primary EWAS approach, which was performed with gee models and included complete DZ twin pairs and one randomly excluded MZ twin for each pair, to (1) an EWAS performed with complete MZ pairs and complete DZ pairs included and (2) an EWAS with a simple linear model (R function lm()) with only one randomly selected twin from each MZ pair and one randomly selected twin from each DZ pair included. In both analyses, the same covariates were included in the model as in the primary EWAS. Third, we repeated the analysis with one randomly selected twin from each MZ pair and one randomly selected twin from each DZ pair without any covariates. Fourth, we performed EWAS analyses in male twins and female twins separately in a simple linear model including only a single randomly selected twin from each pair, again with the same covariates as before (except for sex). Fifth, we repeated the primary EWAS analysis for the DMPs detected in the meta-analysis adjusting, in addition to the same covariates as before, for genotype at the strongest *cis* mQTL SNP of each CpG and three principal components (PCs)

based on the genotype data. One additional comparison was also performed in the Brisbane System Genetics Study (BSGS), which includes 125 MZ twins, 194 DZ twins, and 95 siblings of twins, and 62 parents of twins. In addition to running the EWAS comparing MZ to DZ twins (family members excluded), this cohort also ran an EWAS with MZ twins as cases and DZ twins plus siblings as controls. In sensitivity analyses comparing MZ twins to everyone else (DZ twins, parents, and siblings), DNA methylation $\beta$-value was the outcome variable, and predictors were: MZ twin status (yes/no), sex, age at blood sampling, percentages of monocytes, eosinophils, and neutrophils, BMI, and smoking score. Results of these analyses are described in detail in Supplementary Note 3.

**EWAS meta-analysis.** A P-value-based fixed-effects sample size-weighted meta-analysis was performed in METAL[103]. The sample size-weighted method was chosen because one cohort did not analyze DNA methylation $\beta$-values but residuals (thus the scale of the methylation values was not the same in all cohorts). Only methylation sites that were present in all cohorts were included in the meta-analysis (367,620 methylation sites). Statistical significance was assessed considering Bonferroni correction for the number of sites tested (alpha = 0.05/(367,620 = $1.36 \times 10^{-7}$)). Regional plots were created with coMET[104].

**Twin correlations, cross-tissue correlation, and heritability.** We characterized top sites using previously described[16] estimates of total twin heritability and SNP heritability of DNA methylation in blood, the correlation between longitudinal peripheral blood DNA methylation levels collected with an interval of on average 5 years (based on 31 individuals with 2 longitudinal samples), and the correlation between methylation level in blood and buccal (based on 22 individuals with a blood and buccal sample). In the NTR-ACTION cohort, twin correlations were also computed separately in MZ twin pairs for each chorion type. For each CpG, the Pearson correlation ($r$) was computed between the $\beta$-value of Twin 1 and the $\beta$-value of Twin 2 (across all MZ twin pairs, i.e. MZ twin pairs are cases). These correlations were computed on the residuals derived after adjusting the methylation $\beta$-values for the same set of covariates as included in the EWAS. Absolute within-MZ pair differences in DNA methylation in blood were calculated in NTR on the residuals derived after adjusting the methylation $\beta$-values for the same set of covariates as included in the EWAS, except for age and sex, which are identical in MZ twins. Previously published correlations between DNA methylation levels in the blood and four brain regions from matched samples (prefrontal cortex, entorhinal cortex, superior temporal gyrus, and cerebellum) were obtained from Hannon et al. [17].

**Chorionicity.** The role of chorionicity was examined in the NTR-ACTION cohort. Data on chorionicity were obtained by linking data from the NTR to the Pathologisch Anatomisch Landelijk Geautomatiseerd Archief [Pathological Anatomy National Automatic Archive of the Netherlands] (PALGA) database and biobank[19]. We examined three groups of MZ twins: monochorionic monoamniotic pairs, monochorionic diamniotic pairs, and dichorionic pairs.

**Enrichment analyses.** To examine whether DMPs were enriched near telomeres and centromeres, we made a classification to indicate if a CpG was located within a distance of telomeres or centromeres equivalent to 5% of the total chromosome length (taking 5% on both sides of the centromere, and 5% of both ends of each chromosome). The genomic location enrichment tool from the EWAS atlas[27] was used to test for enrichment of 13 location categories based on the location of CpGs relative to genes and CpG islands. Methylation sites (346) located in DMRs of 59 imprinted genes were obtained from Yuen et al. [21]. Putative metastable epialleles (2210 methylation sites) were obtained from Baak et al. [22]. Age-related variably methylated positions (aVMPs) were obtained from Slieker et al.[26].

To examine the overlap of differentially methylated sites with 15 Epigenomic Roadmap Chromatin States, we used eFORGE V2.0[105]. Transcription factor (TF) motif analysis was performed using eFORGE-TF (https://eforge-tf.altiusinstitute.org/)[105]. Briefly, eFORGE-TF analyses 450k and EPIC array probe sets for overlap enrichment against stringently filtered FIMO TF motif scan matches using motifs from JASPAR, UniPROBE, Taipale, and TRANSFAC databases, adjusting for multiple testing[105]. Next, gene-based pathway analyses were performed on the significant TFs using AmiGO and PANTHER[106–109]. For these analyses, we used the GO Ontology database (released 2020-02-21) and the PANTHER Overrepresentation Test (Released 2020-04-07), with default settings (Fisher's exact test, with annotation set GO biological process complete). We also performed pathway enrichment analysis on the nearest genes of significant DMPs. This analysis was performed in metascape (http://metascape.org)[110] with the nearest genes of all methylation sites that were tested in the meta-analysis as background, species *Homo sapiens*, and all default options otherwise. The trait enrichment tool from the EWAS atlas[27] was used to test for enrichment of methylation sites previously associated with other traits among top sites from the EWAS meta-analysis of zygosity. We tested for enrichment of all traits (488) that were present in the atlas on March 31, 2020.

Two lists of DMPs were used as input for all enrichment analyses: the epigenome-wide significant hypomethylated sites (lower methylation in MZ compared to DZ twins) detected in the meta-analysis (497 DMPs) and the epigenome-wide significant hypermethylated CpGs (higher methylation in MZ

compared to DZ twins) detected in the meta-analysis (337 DMPs). Transcription factor motif analysis was performed using several *p*-value thresholds (results were similar). We present results using a stringent *p*-value cut-off (DMPs with a $p < 5 \times 10^{-18}$; 77 DMPs).

**Methylation QTLs**. Methylation Quantitive Trait Loci (mQTL) results were obtained from the Genetics of DNA Methylation Consortium (GoDMC)[28]. We obtained genome-wide significant results using the significance thresholds as applied by the consortium ($p < 1 \times 10^{-8}$ for *cis* mQTLs and $p < 1 \times 10^{-14}$ for *trans* mQTLs), and performed a look-up of the 834 methylation sites from the EWAS meta-analysis to identify *cis* mQTLs and *trans* mQTLs of these methylation sites.

**Penalized regression models**. Whole blood Illumina 450 K array data from the NTR were used as a training data set to build a DNA methylation-based predictor of "being an MZ twin" with penalized regression models (elastic net; implemented in the R package glmnet[111]). To this end, the NTR data were split at random into a training data set containing 70% of all families and a test data set containing the other 30% (Supplementary Data 12). Two other independent test data sets were considered: BSGS from Australia (blood, 450k array) and the NTR buccal methylation data set from children (EPIC array). Two predictors were trained: Model 1 was trained on data from twins only, to classify their zygosity. Model 2 was trained on data from twins and a small group of family members of twins to distinguish MZ twins from the rest (dizygotic twins and family members).

In the training data, zygosity was regressed on all methylation sites ($N = 381,376$) that (1) were present both on the Illumina 450K and EPIC array and (2) survived quality control in the training set (NTR-blood) and in the test data sets (NTR-buccal, Australia-blood). Second, we tested training on the subset of epigenome-wide significant CpGs from the meta-analysis (833 CpGs that were also present on the EPIC array). The alpha parameter of glmnet was set to 0.5 (elastic net regression) and the lambda value was selected by taking the minimum lambda using 10-fold cross-validation on the training data with the AUC method (R command: cv.glmnet($x$ = methylation, $y$ = zygosity, alpha = 0.5, nfolds = 10, family = "binomial", type.measure = "auc")).

In all training and test data sets, methylation beta values were standardized (z-scores), because preliminary analyses (data not shown) on unstandardized methylation beta values showed weak performance of the binary predictor in test data sets. In the NTR data sets, missing values for probes (probes with missing values in more than 5% of the sample had been removed) were imputed (for penalized regression models only) with the function imputePCA from the package missMDA as implemented in the pipeline for DNA methylation-array analysis developed by the Biobank-based Integrative Omics Study (BIOS) consortium (https://molepi.github.io/DNAmArray_workflow/).

**Reporting summary**. Further information on research design is available in the Nature Research Reporting Summary linked to this article.

## Data availability

The HumanMethylation450 BeadChip data from the NTR are available as part of the Biobank-based Integrative Omics Studies (BIOS) Consortium in the European Genome-phenome Archive (EGA), under the accession code EGAD00010000887. The HumanMethylation450 BeadChip data from E-Risk are accessible from the Gene Expression Omnibus (accession code: GSE105018). The FTC DNA methylation data is deposited in THL Biobank Finland. For information on access and how to apply, see https://thl.fi/en/web/thl-biobank/for-researchers. The applicant for the data can reference the publication when asking for access. The webpage of the THL Biobank describes the exact procedure for accessing the data. The THL biobank grants access to qualified academic and commercial applicants with a scientifically justified study plan. The majority of MZ TwinsUK whole blood DNA methylation profiles are a subset of publicly available data set GEO GSE121633. Additional individual-level data are not permitted to be shared or deposited due to the original consent given at the time of data collection. However, access to these data can be applied for through the TwinsUK data access committee. For information on access and how to apply, see http://www.twinsuk.ac.uk/data-access/submission-procedure/. BSGS DNA methylation data are available at the Gene Expression Omnibus under accession code GSE56105. The NTR-ACTION data sets are available from the Netherlands Twin Register on reasonable request (https://tweelingenregister.vu.nl/information_for_researchers/working-with-ntr-data). Genome-wide summary statistics from the EWAS meta-analysis and weights from the elastic net regression models are provided in Supplementary Data 3, Supplementary Data 13, and 14.

## Code availability

An R-script (EpiPredictorMZtwin.R) and accompanying R data object to apply the epigenetic predictor of MZ twinning is provided in Supplementary Sofware 1. The pipeline for DNA methylation-array analysis developed by the Biobank-based Integrative Omics Study (BIOS) consortium are available here: https://molepi.github.io/DNAmArray_workflow/ (https://doi.org/10.5281/zenodo.3355292). All other analysis code is available upon request from the corresponding author.

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

## Acknowledgements

We acknowledge funding from the Netherlands Organization for Scientific Research (NWO): Biobanking and Biomolecular Research Infrastructure (BBMRI–NL, 184.021.007; 184.033.111), and NWO Large Scale infrastructures, X-Omics (184.034.019). Cohort-specific acknowledgments are provided in Supplementary Note 8. We thank Peter Visscher and Jian Yang for their helpful comments.

## Author contributions

J.v.D.: data analysis, conceived the study, wrote the manuscript with important input from all authors; S.G.: performed epigenome-wide association analysis; performed epigenetic prediction analysis; A.F.M.R.: design, data analysis; V.V.O.: data analysis, figures, literature review; H.M.: data analysis, conceived the study; C.E.B.: performed eFORGE-TF analyses and pathway enrichment analyses of T.F. genes; K.S., S.L., and J.E.C.-F.: performed epigenome-wide association analysis, processed and generated data sets; E.H., F.A.H., C.E.M.v.B., J.J.H., P.-C.T., and E.A.E.: processed and generated data sets; BIOS Consortium: Generated DNA methylation data sets for the Netherlands Twin Register, provided analysis scripts. The Genetics of DNA Methylation Consortium: Performed methylation Q.T.L. analyses and shared mQTL summary statistics for the current study; J.L.M. and G.H.: performed methylation Q.T.L. meta-analyses on data from the GoDMC consortium; F.P., C.D.S., B.T.H., P.E.S., L.D., and S.M.M.: interpretation of results, provided feedback on draft; T.E.M., E.C.J.d.G., G.M., G.W.M., B.R., M.O., J.K., T.D.S., J.T.B., J.M., and A.C.: acquired funding, supervised project activities, provided feedback on the draft. N.G.M.: coordination, contributed to the first draft, D.I.B.: conceived the study, coordination, contributed to the first draft.

## Competing interests

The authors declare no competing interests.

## Additional information

## BIOS Consortium

Jenny van Dongen [1,2,3], Jouke-Jan Hottenga[1,3], Bastiaan T. Heijmans [17], P. Eline Slagboom [17] & Dorret I. Boomsma [1,2,3]

**Genetics of DNA Methylation Consortium**

Jenny van Dongen [1,2,3], Allan F. McRae [5], Karen Sugden[7], Juan E. Castillo-Fernandez [9], Eilis Hannon [10], Terrie E. Moffitt[7,11], Jouke-Jan Hottenga[1,3], Josine L. Min [12,13], Gibran Hemani [12,13], Bastiaan T. Heijmans [17], P. Eline Slagboom [17], Eco J. C. de Geus [1,3], Gonneke Willemsen[1,3], Grant W. Montgomery [5], Miina Ollikainen [8], Jaakko Kaprio [8], Timothy D. Spector[9], Jordana T. Bell [9], Jonathan Mill [10], Avshalom Caspi[7], Nicholas G. Martin [4] & Dorret I. Boomsma [1,2,3]

A full list of members and their affiliations appears in the Supplementary Information.

