## [Peer Review File · Nature Communications]

Title: Identical twins carry a persistent epigenetic signature of early genome programmingREVIEWER COMMENTS

Reviewer #1 (Remarks to the Author):

The paper describes stable methylation signatures in somatic cells among MZ individuals using data from the NTR and replication samples across several other samples from twin registries. Controls primarily included DZ twins, as well as parents and siblings. This study represents the first EWAS analysis to uncover MZ methylation signatures and hence this is novel and of interest to the research community. Indeed, uncovering the signatures may lead to further insights into the rate of MZ twinning in the population.

The authors have conducted thorough analyses and sensitivity analyses that suggest that 243 DMPs were replicated across samples and a combined meta-analysis suggested that 834 CpGs achieved epigenome-wide significance.

The statistical analysis plan was carried out with attention to detail and a common protocol for all replication samples. Others following this study could likely carry out analyses intended to reproduce the work, given the level of detail provided.

Further evidence to strengthen the conclusions include the following --

Lines 88-90: "The number of DMPs that replicated following Bonferroni correction for 234 tests ranged from 5 to 186 (differences likely reflect power)." Should 234243? And can power be further investigated concerning the 5 to 186 replications?

Lines 132-142. The sample sizes concerning the child samples by chorionicity type are small, albeit the findings are compelling. The shift in the correlation distributions is perhaps more compelling than the reported average correlation differences--to what extent is the increasing skew driving the differences?

Concerning the findings that MZ DMPs occur near telomeres and centromeres and polycomb repressed regions characterized by H3K27me3, studies of longevity have reported these and other related pathways may be involved in lifespan regulation. Coupled with findings that MZ twins may have a survival advantage over DZ twins and the general population at most ages in the lifespan (<https://doi.org/10.1371/journal.pone.0171794> 0), to what extent can the current results be explained as differentiating MZ signatures versus longevity-relevant signatures?

Reviewer #2 (Remarks to the Author):

In this manuscript, you conducted a well-powered epigenome-wide association study of MZ twinning (vs DZ twins and DZ twins + singletons). You found strong evidence for an MZ twin-specific signature, although I detail below my easily-addressable criticisms to make the study more robust. You further characterise the genes involved in the signature in the context of an early, shared developmental event at the time of splitting that leaves behind an epigenetic legacy. Your statistical analysis is rigorous and outstanding by EWAS standards. Your results will have a high impact on the field of twin research and will definitely be of interest to a wider audience. I also think that your results will influence the thinking of others in the field.

Points to address

1. Main criticism 1: you have analysed data from singletons, MZ twins and DZ twins. You started by defining MZ twin-specific CpGs by comparing MZ with DZ twins in six cohorts and later replicated in 4 more cohorts and conducted a meta-analysis with all data. However, the manuscript will be greatly improved by comparing MZ twins with the rest of the population e.g. singletons & DZ twins as the primary analysis and performing sensitivity analysis with specific cohorts as the secondary analysis. In addition, a comparison of DZ twins with singletons would provide information on whether there is or isn't a signature common to ALL twins.
2. Main criticism 2: the omission of within-MZ pair data for an epigenomic analysis is a lost opportunity to provide information to support the main analysis. Can you please add in within pair analysis from cohorts for which you have the data? And while you have compared your results that of Waterland's 'supersimilarity' data from MEs within MZ pairs, comparing within-pair results would be a bonus.
3. 'Main', p3, lines 15- 16 "the methylome of the pre-implantation embryo undergoes multiple waves of global DNA demethylation, followed by de novo methylation". I believe that there is only a single round of postzygotic global demethylation; can you please clarify what you mean?
4. Results, p3, line 14 "To examine if MZ twinning is linked to epigenomic changes". I suggest the as this is not a longitudinal study, you should change 'changes' to 'differences'
5. Results, p4, lines 12-14 "Buccal samples consist mainly of epithelial cells (predicted mean=81%), which are derived from the ectodermal cell layer, while white blood cells are derived from the mesodermal lineage." Please supply a reference for this statement.
6. Results, p4, line 17 "The 243 sites..." – as no previous reference to 243 sites, this should be replaced by "Two hundred and forty-three sites..."
7. Results, paragraph spanning pp4-5: The analyses presented in this paragraph on longitudinal analysis, heritability analysis, & cross-tissue comparisons are rather heterogeneous, a little confusing in this format and the rationales for studying each are not clearly articulated. Can you please split to separate paragraphs?
8. Results, p5, lines 6-7 "...had a high total heritability (mean= 57%) and SNP heritability (mean = 14%...": can you please present comparators for those figures i.e. high compared to what?
9. Results, p5, lines 12-19: this text should be in the Discussion
10. Results, p6 lines 7-8 "...multiple CpGs in the protocadherin (PCDH) superfamily gene clusters on chromosome 5q31..." This gene locus contains multiple PCDH genes with very similar sequences. Even though you removed probes shown to cross hybridise by Chen et al, I suggest that you should sound a note of caution in the Discussion about interpreting such results.
11. Results, p6, line 13: "Hypomethylation was enriched near telomeres": can you please define what you mean by "near" in terms of megabasepairs. Same for centromeres. Please also discuss these results in the Discussion.
12. Results, p6: enrichment of DNPs near chromosome landmarks: can you provide statistical backup for their % enrichments i.e. perform a chi squared or similar?
13. Results, p7, lines 8-10: this text should be in the Discussion
14. Results, p8, lines 13-21: this text should be in the Discussion
15. Results optional suggestion: in your datasets, are there sole-survivors of pregnancies that started as twins in which you can test the validity of your twin classifier?
16. Discussion, p9, lines 20-21 "CpG sites in...". Please change to "CpG sites enriched in..."
17. Discussion, p10, lines 1-2 "a phenomenon that shows...". Please change to "a phenomenon that normally shows..."
18. Discussion, p10, final paragraph: please integrate your finding that DMPs are enriched in genes

associated with development and cell fate, and address whether twinning could be classified as a developmental 'abnormality'

19. Discussion general: can you please comment on how your findings will influence the conduct or interpretation of twin EWAS? For example, should subsequent twin EWAS studies be cautious about extrapolation of their findings to singletons?

Jeffrey M Craig, Deakin University, Australia

Reviewer #3 (Remarks to the Author):

Peer Review

Identical twins carry a persistent epigenetic signature of early genome programming

The authors show that MZ twinning is associated with a DNAm signature in somatic tissues (principally blood). They explore genomic locations and features associated with this signature and propose that it can be used to retrospectively diagnose if a person was conceived as a MZ twin.

The methodology used is sound, including use of discovery/replication and meta-analyses, with follow up sensitivity analyses to assess the potential influence of important factors including sex and selection of one or both twins from MZ/DZ pairs. The analysis of the influence of chorionicity on methylation concordance further strengthens evidence that these are genuine associations.

The performance of the penalised regression classifier in identifying MZ twins is surprisingly good. This discovery therefore provides a potentially valuable tool for identifying twins including 'vanishing twins' which would have tremendous benefit for a diverse range of future avenues of research, as suggested by the authors.

I enjoyed reading this paper and thoroughly recommend it for publication. I have made some comments and suggestions below.

MZ twin and zygosity classifiers

This is a novel addition that will be of great interest to the wider research community. However, I think the utility of this discovery as a tool for other researchers could be improved through the following:

1. In ST19 the authors provide AUC for the elastic net MZ classifier, along with sensitivity scores for identification of MZ, DZ and non-twins. It would also be useful to provide data on mis-classification rates for these 3 categories as this will help readers assess its utility for their own research.
2. The authors provide lists of CpGs and coefficients for the zygosity and MZ classifiers in ST20 and ST21. For readers familiar with regression-based classifiers it is reasonably clear how these could be used to make predictions from their own methylation data. However it would be a real benefit for a broader range of readers if the authors could spell out how to use the different classifiers along with the use cases for each. Even better, the authors could provide a very simple R script (or R package) for this purpose.
3. Yet another refinement would be to design a classifier that could distinguish twins (MZ or DZ)

from non-twins. Judging from the results in ST19 it might not be possible to do this reliably, but I would be interested to hear the authors views on this.

4. Finally, it would be helpful for the authors to give their view of the suitability of the classifier(s) for the different use cases outlined above in the manuscript, either in Results or Discussion.

P4 9-11

“The number of DMPs that replicated following Bonferroni correction for 234 tests ranged from 5 to 186 (differences likely reflect power).”

Considering the sample sizes for the different replication cohorts, it's not clear to me how the power statement is justified. For example FTC has similar N (1708) to both the NTR discovery cohort (1957) and E-Risk (1164), but far fewer replicating loci at the Bonferroni threshold. I think further explanation or comment is required on why the number of replicating loci varies so much between cohorts.

Fig 1

Should be 'all other sites are shown in green' (unless there is a problem with colour reproduction in my version)

P5 6-7

“MZ-DMPs showed on average ... a high total heritability (mean= 57%) and SNP heritability (mean = 14%, Fig. S5).”

Some comparison, e.g. with array background or suitable controls is required to substantiate the claim that total and SNP heritabilities are 'high'. The latter in particular may not be justified given the seeming very low heritability (close to zero) indicated by the irregular distribution in Fig S5.

P8 21-25

The observations of trans-mQTL annotated to TRIM28, DNMT3B etc are potentially interesting, but the details on numbers of MZ-DMPs mapping to these are buried in Appendix 5 making it hard to judge their importance. I would suggest adding a little quantitative detail to the main text on this.

P8 25-27 / Methods / Supp Tables 17 & 18

Please explain the meanings of 'DMC' and 'background' in Supp Tables 17 & 18 reporting results of the EWAS atlas used for the EWAS enrichment analysis. Some of the reported odds ratios and enrichment p-values are extremely high / significant so it would be helpful to aid interpretation.

Reviewer #1 (Remarks to the Author):

The paper describes stable methylation signatures in somatic cells among MZ individuals using data from the NTR and replication samples across several other samples from twin registries. Controls primarily included DZ twins, as well as parents and siblings. This study represents the first EWAS analysis to uncover MZ methylation signatures and hence this is novel and of interest to the research community. Indeed, uncovering the signatures may lead to further insights into the rate of MZ twinning in the population.

The authors have conducted thorough analyses and sensitivity analyses that suggest that 243 DMPs were replicated across samples and a combined meta-analysis suggested that 834 CpGs achieved epigenome-wide significance.

The statistical analysis plan was carried out with attention to detail and a common protocol for all replication samples. Others following this study could likely carry out analyses intended to reproduce the work, given the level of detail provided.

* Thank you very much for reviewing our paper. We appreciate that the reviewer recognizes the novelty and value of the work, and acknowledges the value of the common analysis protocol and reproducibility of the analysis.

Further evidence to strengthen the conclusions include the following --

Lines 88-90: "The number of DMPs that replicated following Bonferroni correction for 234 tests ranged from 5 to 186 (differences likely reflect power)." Should 234243? And can power be further investigated concerning the 5 to 186 replications?

*Thank you for spotting this: 234 should indeed be 243. We have corrected this and expanded the section describing the number of DMPs that show replication following stringent Bonferroni correction to provide a broader discussion of likely factors affecting the power to replicate effects in each cohort:

"Since effect sizes were very similar across cohorts (**Figs 1A-D**), differences between cohorts in the number of DMPs that replicated following stringent Bonferroni correction likely reflect power related to the following differences between replication cohorts: total sample size (ranging from 356-1708), zygosity frequencies (ranging from 33% to 80% MZ), and whether correction for inflation of test statistics was required (**Table S3**)."

Further details are given in our response to a related question on this topic from reviewer 3.

Lines 132-142. The sample sizes concerning the child samples by chorionicity type are small, albeit the findings are compelling. The shift in the correlation distributions is perhaps more compelling than the reported average correlation differences--to what extent is the increasing skew driving the differences?

*We agree that the pattern is compelling and added the following sentence to the results section to point out that the small difference in mean is indeed caused by a shift in the

correlation distribution towards a left-skewed distribution that is attributable to a sub-set of CpGs that show a larger correlation in monozygotic MZ twins.

Results: “The distributions of the correlations illustrate that the relatively small mean differences are driven by a sub-set of CpGs that have a larger correlation in monozygotic (especially monozygotic monoamniotic) MZ twins (**Fig. 2c**).”

We agree with the reviewer that the sample size of the chorionicity analyses is a limitation, and have now explicitly mentioned this in the discussion:
“A limitation of our chorionicity analyses is that the sample size was relatively small due to the limited numbers of twins for which reliable chorionicity data and DNA methylation data were available”.

Concerning the findings that MZ DMPs occur near telomeres and centromeres and polycomb repressed regions characterized by H3K27me3, studies of longevity have reported these and other related pathways may be involved in lifespan regulation. Coupled with findings that MZ twins may have a survival advantage over DZ twins and the general population at most ages in the lifespan (<https://doi.org/10.1371/journal.pone.0171794>), to what extent can the current results be explained as differentiating MZ signatures versus longevity-relevant signatures?

*Thank you for this interesting and relevant question. We note that the most recent literature does not support a survival advantage of MZ twins over DZ twins. The paper by Sharrow et al 2016 (PMID: 27192433), mentioned by the reviewer, contained a methodological error (PMID: 2815209). A more thorough re-analysis of the data from the Danish Twin Registry suggested that monozygotic and dizygotic twins have similar lifespans (Hjelmborg 2019 PMID: 30791679).

We do agree with the reviewer that the overlap of MZ-DMPs with ageing-related regions is worth further exploration. We therefore performed additional analyses to explicitly test for enrichment of ageing-associated CpGs among MZ-DMPs and expanded the results and discussion section to describe the overlap of our findings with ageing-associated DNA methylation signatures. In the previous version of the manuscript, enrichment analyses of age-DMPs (CpGs whose mean level correlates with age) were already performed as part of the analyses in which we tested for enrichment of all traits (>400) reported in the EWAS atlas. We now additionally added an additional enrichment analysis for age-VMPs (CpGs whose variance correlates with age).

The results section now contains a new sub-section “ageing”:

“Ageing

Several of the regions that show enrichment among MZ-DMPs, including sub-telomeric regions²⁴, Polycomb-repressed regions²⁵, and the protocadherin gene clusters²⁶ have also been associated with longevity or ageing. We therefore looked in more detail at the overlap of age-related DNA methylation variation and MZ-DMPs. We tested for enrichment of methylation sites previously associated with > 400 traits reported in the EWAS atlas²⁷ (**Appendix 5, Tables S15, S16**) including age-DMPs (i.e. CpGs whose mean methylation level correlates with age) from a number of studies. In addition, we

tested for enrichment of age-VMPs (i.e. CpGs whose variance correlates positively with age)²⁶. While age-DMPs were not enriched among MZ-DMPs (**Tables S15, S16**), age-VMPs were significantly enriched among both MZ-hypo and MZ-hyper DMPs (**Tables S17, S18**). “

We also added the following sentences to the discussion:

“We note that sub-telomeric regions¹, as well as several other regions that show enrichment among MZ-DMPs, namely, Polycomb-repressed regions², and the protocadherin gene clusters³ have been previously associated with longevity or ageing. Furthermore, we observed that age-variable methylated positions, i.e. CpGs whose variance correlates positively with age, were significantly enriched among MZ-DMPs. We interpret the overlap with ageing-related loci as an indication that MZ twinning and age-related epigenetic dysregulation both affect loci whose methylation pattern is established early in development. We note that the observation is unlikely to have implications for the lifespan of MZ twins, because data from the largest population-based and oldest twin cohort ever studied (109,303 twins from the Danish Twin Registry born between 1870 and 1990) concluded that monozygotic and dizygotic twins have similar lifespans⁴.”

Reviewer #2 (Remarks to the Author):

In this manuscript, you conducted a well-powered epigenome-wide association study of MZ twinning (vs DZ twins and DZ twins + singletons). You found strong evidence for an MZ twin-specific signature, although I detail below my easily-addressable criticisms to make the study more robust. You further characterise the genes involved in the signature in the context of an early, shared developmental event at the time of splitting that leaves behind an epigenetic legacy. Your statistical analysis is rigorous and outstanding by EWAS standards. Your results will have a high impact on the field of twin research and will definitely be of interest to a wider audience. I also think that your results will influence the thinking of others in the field.

*Thank you very much for reviewing our paper. We are delighted to read that you recognize the value and quality of our study.

Points to address

1. Main criticism 1: you have analysed data from singletons, MZ twins and DZ twins. You started by defining MZ twin-specific CpGs by comparing MZ with DZ twins in six cohorts and later replicated in 4 more cohorts and conducted a meta-analysis with all data. However, the manuscript will be greatly improved by comparing MZ twins with the rest of the population e.g. singletons & DZ twins as the primary analysis and performing sensitivity analysis with specific cohorts as the secondary analysis. In addition, a comparison of DZ twins with singletons would provide information on whether there is or isn't a signature common to ALL twins.

*Thank you for this comment, which made us realize that some of the analyses you suggest, which we had already performed, were hidden in the supplemental material, and that we failed to explain the rationale for our analytical approach. Our main

EWAS analysis was a comparison of MZ versus DZ twins. DZ twins represent the ideal control group for detecting a DNA methylation signature of MZ twinning because DZ twins, like MZ twins (but unlike singletons), experience the unique prenatal condition of sharing a womb with a co-twin, thus by having DZ twins (rather than singletons) as a control group, we control for possible effects of sharing a womb with a co-twin. We performed sensitivity analyses in which we compared 1) MZ twins to singletons and 2) DZ twins to singletons (we apologize that these analyses were hidden in the supplement – appendix 3).

Results from these sensitivity analyses ruled out that results from the main analysis are attributable to a DZ-specific or twin-specific methylation signature but rather indicated the existence of an MZ-specific signature. We wish to note that the amount of data available for singletons was relatively small: DNA methylation data from singletons was only available in the Netherlands Twin Register (NTR) and the Brisbane Systems Genetics Study (BSGS, see Table 1). Therefore, sensitivity analyses (i.e. comparisons to singletons) were performed in the NTR and BSGS. The comparison of MZ twins versus all others (DZ twins+non-twins) that you suggest yielded highly similar results as the main analysis of MZ versus DZ twins (Appendix 3, Table S2, Figure S2B), and we therefore do not expect that adding the singletons to the control group from the main analyses would make a notable difference to the results. This is further corroborated by the results from our elastic net regression models, where we tested the performance of two models: 1) we trained the model on MZ and DZ twin data only, 2) we trained the model to distinguish MZ twins from the rest of the population (=DZ twins+singletons). The performance of these models was very similar. To clarify our analytical approach, and to put more emphasis on analyses previously hidden in the supplement, we have added/modified the following sentences in the results section of the main text:

“DZ twins represent the ideal control group for detecting a DNA methylation signature of MZ twinning because DZ twins, like MZ twins (but unlike singletons), experience the unique prenatal condition of sharing a womb with a co-twin, thus controlling for possible effects of sharing a womb with a co-twin.”

“Sensitivity analyses were conducted in the NTR discovery cohort and in Brisbane Systems Genetics Study (BSGS), because these cohorts also had DNA methylation data available for non-twins (**Table 1**).”

“By contrast, a comparison of DZ twins to non-twins yielded no epigenome-wide significant DMPs, and showed no strong concordance of effect sizes compared to the main (MZ vs DZ twin) analysis (**Table S2, Fig. S1**), indicating that the results from our primary EWAS (mainly) reflect differential DNA methylation in MZ twins.”

2. Main criticism 2: the omission of within-MZ pair data for an epigenomic analysis is a lost opportunity to provide information to support the main analysis. Can you please add in within pair analysis from cohorts for which you have the data? And while you have compared your results that of Waterland's 'supersimilarity' data from MEs within MZ pairs, comparing within-pair results would be a bonus.

* Thank you for this interesting suggestion. We previously reported the MZ twin correlations as a measure of co-twin similarity, but we now also report within-pair DNA methylation differences.

Results:

“Most notable is the pattern of twin correlations in blood (Fig. S6), with MZ twin correlations of the methylation level of the 834 sites being almost three times larger on average compared to DZ twin correlations (mean MZ correlation=0.58, mean DZ correlation=0.20). In line with the moderate to large MZ twin correlations at MZ DMPs, histograms of within-pair differences (Fig. S7) and scatterplots (twin 1 versus twin 2) of methylation levels in MZ twin pairs (Fig. S8) illustrate that at each MZ-DMP, most MZ pairs have highly concordant methylation levels. Mean absolute within-MZ pair differences in DNA methylation level ranged from 0.8% to 7.8% for different MZ-DMPs (mean=3.6%). MZ-DMPs with a larger mean difference between MZ and DZ twins also tended to display a larger mean MZ within-pair difference ($r=0.69$, $p < 2.2 \times 10^{-16}$, Fig. S9). Within-pair differences at MZ-DMPs typically showed wide distributions (Fig. S10), which illustrates that each CpG displayed more pronounced differences in a subset of MZ pairs. Similarly, each MZ pair showed large within-pair differences at a subset of MZ-DMPs.”

3. ‘Main’, p3, lines 15- 16 “the methylome of the pre-implantation embryo undergoes multiple waves of global DNA demethylation, followed by de novo methylation”. I believe that there is only a single round of postzygotic global demethylation; can you please clarify what you mean?

*Latest insights on preimplantation DNA methylation dynamics based on single-cell DNA methylome sequencing of human embryos suggest that there are 3 waves of postzygotic DNA demethylation and 2 waves of postzygotic *de novo* methylation. This is reported in the article cited in this sentence (Zhu et al 2018). This article showed that the first wave occurred during the first 10 to 12h after fertilization. The second wave of global demethylation occurred from the late zygote to the two-cell stage. The third wave occurred from the eight-cell to the morula stage. The first wave of *de novo* DNA methylation occurred from the early male pronuclear to the mid-pronuclear stage and the second occurred from the four-cell to the eight-cell stage.

4. Results, p3, line 14 “To examine if MZ twinning is linked to epigenomic changes”. I suggest the as this is not a longitudinal study, you should change ‘changes’ to ‘differences’

*Thank you for pointing this out. We have replaced “changes” for “profiles”.

5. Results, p4, lines 12-14 “Buccal samples consist mainly of epithelial cells (predicted mean=81%), which are derived from the ectodermal cell layer, while white blood cells are derived from the mesodermal lineage.” Please supply a reference for this statement.

*We’ve added two references (Theda et al 2018, van Dongen et al 2018), and clarified that the 81% cited refers to the NTR child cohort referred to in the preceding sentence:

“Buccal samples consist mainly of epithelial cells^{5,6} (predicted mean in the NTR child cohort=81%), which are derived from the ectodermal cell layer, while white blood cells are derived from the mesodermal lineage.”

14. Theda, C. *et al.* Quantitation of the cellular content of saliva and buccal swab samples. *Sci. Rep.* **8**, 6944 (2018).
15. van Dongen, J. *et al.* Genome-wide analysis of DNA methylation in buccal cells: a study of monozygotic twins and mQTLs. *Epigenetics Chromatin* **11**, 54 (2018).

6. Results, p4, line 17 “The 243 sites...” – as no previous reference to 243 sites, this should be replaced by “Two hundred and forty-three sites...”

*We currently followed the rule to write numbers between zero and ten as words and numbers greater than ten as numbers (e.g. 243) and are happy to make any stylistic changes that the editorial team deems necessary to adhere to the journal’s guidelines.

7. Results, paragraph spanning pp4-5: The analyses presented in this paragraph on longitudinal analysis, heritability analysis, & cross-tissue comparisons are rather heterogeneous, a little confusing in this format and the rationales for studying each are not clearly articulated. Can you please split to separate paragraphs?

*We agree that this paragraph covered many topics and followed the suggestion to split it into separate paragraphs.

8. Results, p5, lines 6-7 “...had a high total heritability (mean= 57%) and SNP heritability (mean = 14%...”: can you please present comparators for those figures i.e. high compared to what?

*Thank you for pointing out that this was unclear. We meant that the estimates are high in comparison to all genome-wide methylation sites. We have clarified the sentence as follows:

“In comparison to genome-wide methylation sites, MZ-DMPs had a high total heritability (mean heritability MZ-DMPs=57%, mean heritability genome-wide autosomal methylation sites= 19%⁷) and SNP heritability (mean SNP heritability MZ-DMPs= 14%, mean SNP heritability genome-wide autosomal sites= 7%⁷; **Fig. S5**)”.

9. Results, p5, lines 12-19: this text should be in the Discussion

*Thank you for this suggestion. We have moved this text to the discussion.

10. Results, p6 lines 7-8 “...multiple CpGs in the protocadherin (PCDH) superfamily gene clusters on chromosome 5q31...” This gene locus contains multiple PCDH genes with very similar sequences. Even though you removed probes shown to cross hybridise by Chen et al, I suggest that you should sound a note of caution in the Discussion about interpreting such results.

*Thank you for this important comment. We indeed removed cross-hybridizing probes according to the most commonly used definition based on an overlap of ≥ 47 bases per probe. In response to your comment, we examined to what extent cross-hybridization due to lower levels of sequence overlap of probes (PMID: 27924034, PMID: 33554115) could have affected our findings. First, we examined how many MZ-DMPs would be excluded if we would apply a more stringent filter to remove potentially cross-hybridizing probes, based definition of an overlap ≥ 30 bases per probe. Second, we evaluated the sequence similarity of the probes underlying our 834 MZ-DMPs. We have added the results to **Appendix 4**, and added a few lines to the discussion.

Appendix 4

“We note that correlations between methylation levels at different CpGs may also arise due to cross-hybridization of probes to multiple locations. We note that we already excluded probes reported by Chen *et al* with an overlap of at least 47 bases per probe¹, which is the most commonly used exclusion criterium in EWA studies, from all of our analyses to avoid this issue. We additionally examined a more stringent definition based on a lower degree of sequence overlap of 30 bases per probe^{1,2}, which flagged 18 of the 834 MZ-DMPs (2.1%) as potentially cross-hybridizing. We have flagged these DMPs in **Table S4**.

The sequence similarity of probes for the 834 MZ-DMPs was generally low. On average, 3.5 bases overlapped, the maximum overlap was 26 bases, and 685 CpGs (82%) are targeted by probes that show less than 14 bases overlap with probes for other MZ-DMPs. We examined one region in more detail; the *PCDH* gene clusters on chromosome 5, because the genes in this region are known to show large sequence similarity. Our EWAS meta-analysis identified 79 MZ-DMPs in this region (**Fig. S3B**). Among the 79 CpGs, the overlap in probe sequences between probes for different CpGs was on average only 3.5 bases, the maximum overlap was 21 bases and 77 of the 79 CpGs had less than 14 overlapping bases. This illustrates that the probes for these 79 CpGs are designed to target largely distinct sequences within the *PCDH* gene clusters, however, we note that all 79 CpGs are targeted by probes that show a small degree of off-target sequence overlap (≥ 14 bases) with other sequences within this genomic region.

¹Chen, Y. *et al*. Discovery of cross-reactive probes and polymorphic CpGs in the Illumina Infinium HumanMethylation450 microarray. *Epigenetics* **2294**, (2013).

²Zhou, W., Laird, P. W. & Shen, H. Comprehensive characterization, annotation and innovative use of Infinium DNA methylation BeadChip probes. *Nucleic Acids Res.* (2017). doi:10.1093/nar/gkw967

“

Discussion:

“We reported some loci with methylation differences extending across large regions of correlated CpGs (such as the *PCDH* superfamily gene clusters), but note that this finding should be interpreted with some caution, because correlations between methylation levels at different CpGs may also arise due to underlying DNA sequence similarity. We excluded probes affected by SNPs and cross-reactive probes (probes that are predicted to hybridize to multiple locations in the genome based on large sequence similarity) from all analyses, as is common practice in EWA studies, and flagged a small number of MZ-DMPs that are measured by probes with a smaller degree of sequence similarity to multiple regions. We note that cross-hybridization arising from even a very small degree of probe off-target similarity (i.e. 14 bases) can affect EWAS results in certain cases, namely when the phenotype studied is associated with a repeat expansion⁴⁹. The extent to which EWA studies in general, including ours, are affected by such low levels of probe overlap with other genomic sequences is unknown. We note that excluding probes with some off-target sequence is undesirable as it would lead to the removal of practically all probes⁴⁹. Rather, future studies with sequencing-based techniques could provide insight into the methylation signature of MZ twins at full resolution, and address the potential contribution of underlying DNA sequence variation.”

Methods

“The following probes were removed: and ambiguous mapping probes reported by Chen *et al* with an overlap of at least 47 bases per probe⁶². In addition, we examined and flagged potentially cross-hybridizing probes at lower levels of sequence overlap (**Appendix 4**), and examined the sequence similarity of probes underlying significant MZ-DMPs detected in our meta-analysis with the R package DNAmCrosshyb (https://github.com/pjhop/dnamarray_crossreactivity⁴⁹). We considered two previously described stringent thresholds of overlap: ≥ 30 bases & ≥ 14 bases⁴⁹”.

11. Results, p6, line 13: “Hypomethylation was enriched near telomeres”: can you please define what you mean by “near” in terms of megabasepairs. Same for centromeres. Please also discuss these results in the Discussion.

*Thank you for this suggestion. “Near telomeres” means that the CpG is located within 5% of the chromosome end, and “near centromeres” means that the CpG is located in the 5% of a chromosome adjacent to the centromere. This information was previously only provided in the methods section. We have now added it to the results section:

“To examine whether DMPs were enriched near telomeres and centromeres, we made a classification to indicate if a CpG was located within a distance of telomeres or centromeres equivalent to 5% of the total chromosome length.”

We have now also added a section to the discussion where we discuss these results (see also our reply to reviewer 1).

“MZ-DMPs were strongly enriched near telomeres and centromeres; but the reason for this remains to be established. We note that sub-telomeric regions¹, as well as several other regions that show enrichment among MZ-DMPs, namely, Polycomb-repressed regions², and the protocadherin gene clusters³ have been previously associated with longevity or ageing. Furthermore, we observed that age-variable methylated positions, i.e. CpGs whose variance correlates positively with age, were significantly enriched among MZ-DMPs. We interpret the overlap with ageing-related loci as an indication that MZ twinning and age-related epigenetic dysregulation both affect loci whose methylation pattern is established early in development. We note that the observation is unlikely to have implications for the lifespan of MZ twins: The most thorough analysis to date, on data from the largest population-based and oldest twin cohort ever studied (109,303 twins from the Danish Twin Registry born between 1870 and 1990) concluded that monozygotic and dizygotic twins have similar lifespans⁴. “

12. Results, p6: enrichment of DNPs near chromosome landmarks: can you provide statistical backup for their % enrichments i.e. perform a chi squared or similar?

* Statistical tests have been performed for all enrichment analyses. These results are provided in the supplemental tables, and in the case of the enrichment analysis of chromatin states in supplemental figure S12 and S13.

13. Results, p7, lines 8-10: this text should be in the Discussion

*We moved these lines to the discussion.

14. Results, p8, lines 13-21: this text should be in the Discussion

*We moved these lines to the discussion.

15. Results optional suggestion: in your datasets, are there sole-survivors of pregnancies that started as twins in which you can test the validity of your twin classifier?

*We agree that this would be a valuable analysis, but unfortunately such information is not available in our datasets and we are not aware of any existing DNA methylation dataset where such information is available.

16. Discussion, p9, lines 20-21 “CpG sites in...”. Please change to “CpG sites enriched in...”

*Done.

17. Discussion, p10, lines 1-2 “a phenomenon that shows...”. Please change to “a phenomenon that normally shows...”

*Thanks for pointing this out, we changed to “a phenomenon that typically shows”.

18. Discussion, p10, final paragraph: please integrate your finding that DMPs are enriched in genes associated with development and cell fate, and address whether twinning could be classified as a developmental ‘abnormality’

*In our view, MZ twinning in humans could indeed be regarded as a developmental ‘abnormality’, judging by the very nature of the MZ twinning event (single births are the norm in humans), and the fact that MZ twin pregnancies are a risk factor for complications. But we also feel that we cannot draw such conclusions based on the current association results that cannot distinguish cause and effect. We note that further functional studies are warranted to delineate mechanisms.

19. Discussion general: can you please comment on how your findings will influence the conduct or interpretation of twin EWAS? For example, should subsequent twin EWAS studies be cautious about extrapolation of their findings to singletons?

*Thank you for this important question. To address this issue, we’ve added the following paragraph to the discussion

“A question that follows from our findings is whether there might be implications epigenetic studies in twins in general. To consider this question, it is important to note that the 834 MZ differentially methylated sites represent only a tiny fraction of all genome-wide sites, although it is unknown whether more differentially methylated loci might exist that may be uncovered by future larger EWA studies, or by other techniques with greater coverage (e.g. bisulfite sequencing). Second, whether any of the methylation differences have phenotypic consequences for MZ twins is unknown. We do not foresee these findings to impact the generalizability of (discordant) MZ twin studies that aim to detect epigenetic variation connected to a trait or disease, unless the trait is connected to the MZ twinning process itself. The literature suggests that for most outcomes studied, MZ twins are very comparable to non-twins, except for some congenital disorders, birth weight and traits strongly related to birth weight⁴⁹. MZ-DMPs showed MZ twin correlations that were on average more than three times as large as the DZ twin correlations, which can indicate a strong genetic influence (i.e. DNA sequence effect) on these DNA methylation sites or be indicative of mitotic inheritance of a pre-twinning established methylation state. The latter scenario has implications for the interpretation of classical twin studies of epigenetic marks in which correlation patterns of MZ and DZ

twins are contrasted to estimate the heritability of epigenetic marks (but as noted, this remark only applies to a limited set of loci in the genome).”

The literature suggests that for most outcomes studied, MZ twins are very comparable to non-twins, Jeffrey M Craig, Deakin University, Australia

Reviewer #3 (Remarks to the Author):

Peer Review

Identical twins carry a persistent epigenetic signature of early genome programming

The authors show that MZ twinning is associated with a DNAm signature in somatic tissues (principally blood). They explore genomic locations and features associated with this signature and propose that it can be used to retrospectively diagnose if a person was conceived as a MZ twin.

The methodology used is sound, including use of discovery/replication and meta-analyses, with follow up sensitivity analyses to assess the potential influence of important factors including sex and selection of one or both twins from MZ/DZ pairs. The analysis of the influence of chorionicity on methylation concordance further strengthens evidence that these are genuine associations.

The performance of the penalised regression classifier in identifying MZ twins is surprisingly good. This discovery therefore provides a potentially valuable tool for identifying twins including ‘vanishing twins’ which would have tremendous benefit for a diverse range of future avenues of research, as suggested by the authors.

I enjoyed reading this paper and thoroughly recommend it for publication. I have made some comments and suggestions below.

*Thank you very much for the positive assessment of our paper and your suggestions for improvement.

MZ twin and zygosity classifiers

This is a novel addition that will be of great interest to the wider research community. However, I think the utility of this discovery as a tool for other researchers could be improved through the following:

1. In ST19 the authors provide AUC for the elastic net MZ classifier, along with sensitivity scores for identification of MZ, DZ and non-twins. It would also be useful to provide data on mis-classification rates for these 3 categories as this will help readers assess its utility for their own research.

*Thank you for this suggestion. We've added the misclassification rates to ST19.

2. The authors provide lists of CpGs and coefficients for the zygosity and MZ classifiers in ST20 and ST21. For readers familiar with regression-based classifiers it is reasonably clear how these could be used to make predictions from their own methylation data. However it would be a real benefit for a broader range of readers if the authors could

spell out how to use the different classifiers along with the use cases for each. Even better, the authors could provide a very simple R script (or R package) for this purpose. *Thank you for this suggestion. We have added R-code to run the best-performing predictor, and added this information to the data availability statement and to the legend of supplemental table S19.

Updated data availability statement:

“An R-script (EpiPredictorMZtwin.R) and accompanying R data object to apply the epigenetic predictor of MZ twinning is provided in data1.”

3. Yet another refinement would be to design a classifier that could distinguish twins (MZ or DZ) from non-twins. Judging from the results in ST19 it might not be possible to do this reliably, but I would be interested to hear the authors views on this.

*We indeed do not expect that we could build a reliable classifier to distinguish twins from non-twins. We did not perform this analysis because our EWAS results convincingly pointed to a strong MZ twin DNA methylation signature, while the EWAS comparing DZ twins to non-twins did not reveal any signal (see response to comment 1 from reviewer 2). We further note the relatively small number of matched non-twin samples (Table 1) .

4. Finally, it would be helpful for the authors to give their view of the suitability of the classifier(s) for the different use cases outlined above in the manuscript, either in Results or Discussion.

* We have now emphasized the lack of signal in the comparison of DZ twins versus non-twins in the results section:

“By contrast, a comparison of DZ twins to non-twins yielded no epigenome-wide significant DMPs, and showed no strong concordance of effect sizes compared to the main (MZ vs DZ twin) analysis (**Table S2, Fig. S1**), indicating that the results from our primary EWAS (mainly) reflect differential DNA methylation in MZ twins.”

Additionally, we have now clarified in the results section that we examined the performance of two MZ twin predictors (one trained on MZ vs DZ twin data, and one trained on MZ versus DZ twin+non-twin data), but that the performance of these predictors was very similar (one performed slightly better in one test dataset, while the other performed slightly better in another test dataset). We apologize that this information was previously hidden in the supplementary material.

“We compared models based on two input sets (genome-wide methylation sites versus meta-analysis DMPs), and trained on two phenotypes (MZ versus DZ twins, and MZ twins versus everyone else (including DZ twins and family members of twins). Predictors trained on the smaller input set of meta-analysis DMPs performed better compared to predictors trained on genome-wide sites, but whether we trained on MZ versus DZ twins or on MZ versus everyone else had little impact on the performance.”

P4 9-11

“The number of DMPs that replicated following Bonferroni correction for 234 tests ranged from 5 to 186 (differences likely reflect power).”

Considering the sample sizes for the different replication cohorts, it's not clear to me how the power statement is justified. For example FTC has similar N (1708) to both the NTR

discovery cohort (1957) and E-Risk (1164), but far fewer replicating loci at the Bonferroni threshold. I think further explanation or comment is required on why the number of replicating loci varies so much between cohorts.

* We have expanded the section describing the number of DMPs that show replication following stringent Bonferroni correction to provide a broader discussion of likely factors affecting the power to replicate effects in each cohort:

“Since effect sizes were very similar across cohorts (**Figs 1A-D**), differences between cohorts in the number of DMPs that replicated following stringent Bonferroni correction likely reflect power related to the following differences between replication cohorts: total sample size (ranging from 356-1708), zygosity frequencies (ranging from 33% to 80% MZ), and whether correction for inflation of test statistics was required (**Table S3**).”

Please note that although the number of replicating CpGs is very large in E-Risk and much smaller in FTC following stringent Bonferroni correction, effect sizes are very similar to effect sizes in the discovery cohort (NTR) in both cohorts (as illustrated by the scatterplots in fig 1A and 1B). The difference in this case was caused because of the need to adjust for genome-wide test statistic inflation in FTC. Adjusting for genome-wide test statistic inflation is the most robust approach to avoid false-positives but it comes at the cost of reducing the power to detect true positives.

Fig 1

Should be ‘all other sites are shown in green’ (unless there is a problem with colour reproduction in my version)

*Thank you for noticing this error. We have corrected it.

P5 6-7

“MZ-DMPs showed on average ... a high total heritability (mean= 57%) and SNP heritability (mean = 14%, Fig. S5).”

Some comparison, e.g. with array background or suitable controls is required to substantiate the claim that total and SNP heritabilities are ‘high’. The latter in particular may not be justified given the seeming very low heritability (close to zero) indicated by the irregular distribution in Fig S5.

* We have now clarified this sentence, as indicated in response to comment 8 from reviewer 2. The reviewer is right that many sites have a SNP heritability close to zero, however, in comparison to the average SNP heritability of all genome-wide methylation sites, the average SNP heritability of MZ-DMPs is substantial (it is more than twice as large on average).

P8 21-25

The observations of trans-mQTL annotated to TRIM28, DNMT3B etc are potentially interesting, but the details on numbers of MZ-DMPs mapping to these are buried in Appendix 5 making it hard to judge their importance. I would suggest adding a little quantitative detail to the main text on this.

*We thank the reviewer for this suggestion and have now moved this information from Appendix 5 to the results section of the main text.

P8 25-27 / Methods / Supp Tables 17 & 18

Please explain the meanings of 'DMC' and 'background' in Supp Tables 17 & 18 reporting results of the EWAS atlas used for the EWAS enrichment analysis. Some of the reported odds ratios and enrichment p-values are extremely high / significant so it would be helpful to aid interpretation.

*We apologize for the omission to clarify these terms. We've now added the meaning of these terms to the legends of these tables.

DMC=Differentially methylated CpGs - This is the number of methylation sites that is an MZ-DMP and has also been previously associated with the trait reported in the first column.

Background= Total number of methylation sites previously associated with the trait in the first column.

REVIEWERS' COMMENTS

Reviewer #1 (Remarks to the Author):

The authors were very responsive to comments in their revisions.

I appreciated the revisions related to power, the exploration of aging-related regions, and the citation on the reanalysis regarding longevity and zygosity (affirming the similarity of MZs and DZs). In the present investigation of aging-related regions, that the 'CpGs whose variance correlates positively with age were significantly enriched among MZ-DMPs' is interesting. Age-related CPGs appear to be among the more heritable sites among aging twins compared to background CpGs [c.f., PMC7431820] and these MZ-DMPs more so at on average 57%.

Reviewer #2 (Remarks to the Author):

Thank you for taking the time to write a detailed reply to all my comments. I am satisfied that you have addressed them all.

Reviewer #3 (Remarks to the Author):

Thank you to the authors for fully addressing my comments and suggestions. I am happy to recommend the manuscript for publication.

We are happy to see that the reviewers are satisfied with the revisions. We very much appreciated their feedback on the manuscript and would like to thank them once again for their time and valuable comments.